# Machine learning-guided deconvolution of plasma protein levels

Maik Pietzner [1,2,3 ✉], Carl Beuchel [1,3], Kamil Demircan[1,2], Julian Hoffmann Anton[2], Wenhuan Zeng[1], Werner Römisch-Margl[4], Summaira Yasmeen [1], Burulça Uluvar [1], Martijn Zoodsma[1,3], Mine Koprulu[2], Gabi Kastenmüller[4], Julia Carrasco-Zanini[2] & Claudia Langenberg [1,2,3 ✉]

## Abstract

Proteomic techniques now measure thousands of proteins circulating in blood at population scale, but successful translation into clinically useful protein biomarkers is hampered by our limited understanding of their origins. Here, we use machine learning to systematically identify a median of 20 factors (range: 1-37) out of >1800 participant and sample charateristics that jointly explained an average of 19.4% (max. 100.0%) of the variance in plasma levels of ~3000 protein targets among 43,240 individuals. Proteins segregated into distinct clusters according to their explanatory factors, with modifiable characteristics explaining more variance compared to genetic variation (median: 10.0% vs 3.9%), and factors being largely consistent across the sexes and ancestral groups. We establish a knowledge graph that integrates our findings with genetic studies and drug characteristics to guide identification of potential drug target engagement markers. We demonstrate the value of our resource by identifying disease-specific biomarkers, like matrix metalloproteinase 12 for abdominal aortic aneurysm, and by developing a widely applicable framework for phenotype enrichment (R package: https://github.com/comp-med/r-prodente). All results are explorable via an interactive web portal (https://omicscience.org/apps/prot_foundation).

**Keywords** Plasma Proteomics; Biomarker; Drugs; Enrichment
**Subject Categories** Computational Biology; Methods & Resources; Proteomics

## Introduction

High-throughput plasma proteomics is now fuelling a new wave of biomarker studies at unprecedented scale (Topol, 2024), but is about to repeat the failures of decades of previous studies with very few, if any, of thousands of tested candidates ever improving clinical practice (Ioannidis and Bossuyt, 2017). The ever-expanding content of proteomic assays, now surpassing half of the protein-coding genome, further exceeds our ability to understand the origins and relevance of the many proteins that are detectable but have no established role in blood (Fig. 1A) (Uhlén et al, 2015, 2019).

Proteome-wide genome-wide association studies (Sun et al, 2023; Dhindsa et al, 2023; Eldjarn et al, 2023; Pietzner et al, 2021; Sun et al, 2018; Suhre et al, 2021; Emilsson et al, 2018) provided evidence that changes in plasma protein abundance can reflect altered production in tissues and identified at least one protein quantitative trait locus (pQTL) for most protein targets. Some pQTLs explained jointly as much as 70% of the variance in plasma levels (Sun et al, 2023; Pietzner et al, 2021) and they have been widely advocated as instruments for causal inference or even to impute the plasma proteome based on comparatively cheap genotyping (Zheng et al, 2020; Zuber et al, 2022; Xu et al, 2023). However, on average, pQTLs explain relatively little of protein variation in plasma, partly because static germline genetic variation does not capture dynamic adaptations that indicate early disease states or worsening of conditions. Atlas-like efforts are now emerging that statistically associate changes in plasma protein levels with hundreds of diseases, ageing clocks, or health characteristics (Deng et al, 2024; Garcia et al, 2024; Eldjarn et al, 2023; Meyer and Schumacher, 2024), but that do rarely translate into biological knowledge or understanding of the factors that underly disease associations or predictive models.

Here, we present a framework for the integration of multimodal data to systematically identify factors, including characterisitics of human health and disease but also technical measures, explaining variation in plasma protein levels building on our previous work (Carrasco-Zanini et al, 2024b). We demonstrate that a relatively small number of factors (median: 20; range: 1–37) of all >1800 tested explain a considerable part of protein variation (>25% for most protein targets) in plasma. Protein targets thereby segregated into distinct clusters explained by indicators of human health but also pre-analytical variation, such as accidental activation of platelets, and identify proteins that are best explained by different characteristics across the sexes and ancestral strata. We demonstrate how the collective knowledge generated here integrated with human genetic studies can guide the identification of disease-

[1]Computational Medicine, Berlin Institute of Health at Charité – Universitätsmedizin Berlin, Berlin, Germany. [2]Precision Healthcare University Research Institute, Queen Mary University of London, London, UK. [3]DZHK (German Centre for Cardiovascular Research), partner site Berlin, Berlin, Germany. [4]Institute of Computational Biology, Helmholtz Zentrum München - German Research Center for Environmental Health, Neuherberg, Germany. ✉E-mail: maik.pietzner@bih-charite.de; claudia.langenberg@qmul.ac.uk

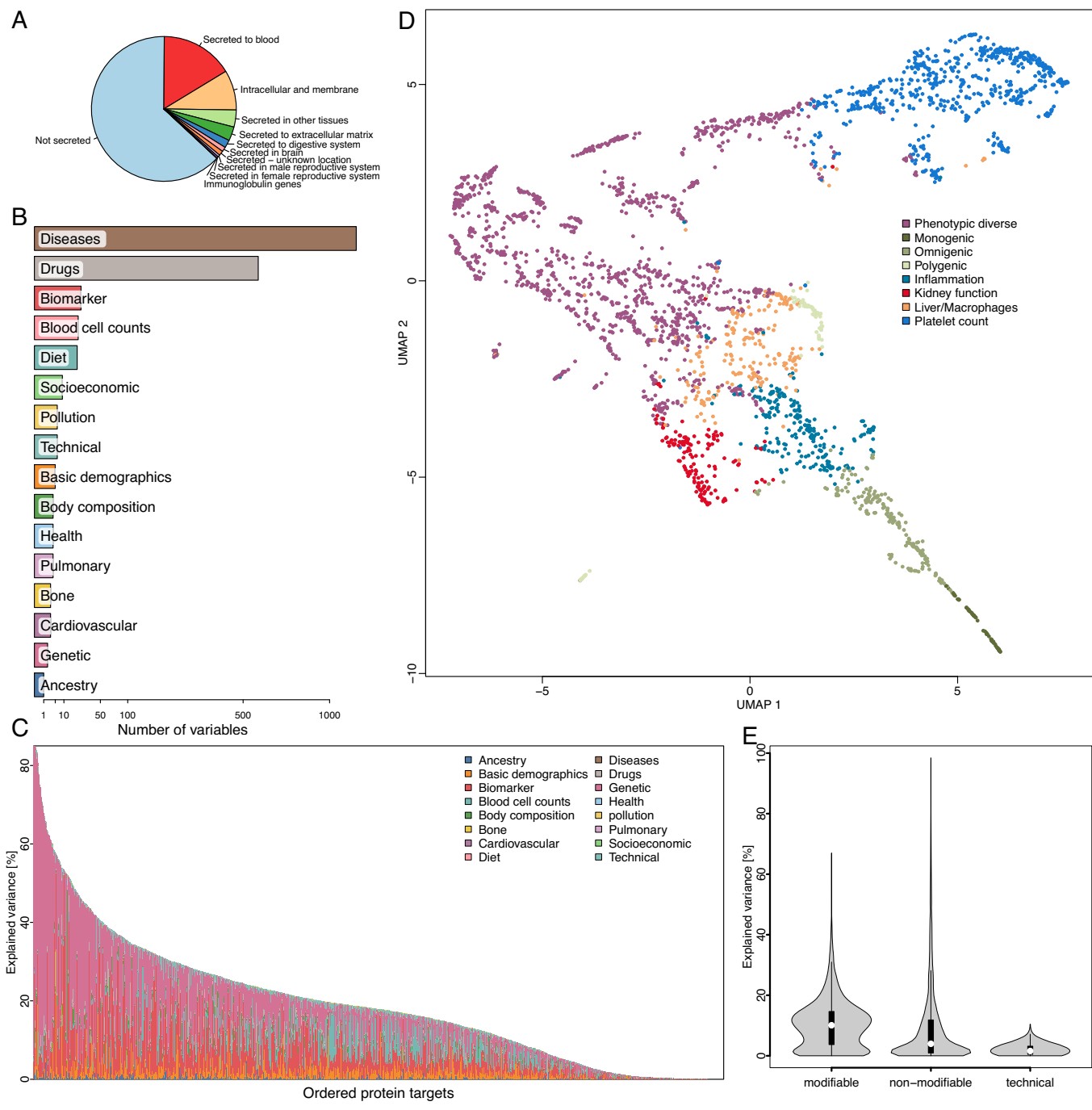

**Figure 1. A coordinate system of plasma protein variation.**

(A) Secretome assignment (Uhlén et al, 2019) for 2919 protein targets analysed in the present study. (B) Number and type of UK Biobank participant characteristics considered in the study. (C) Stacked bar charts displaying the achieved variance explained for each protein target. Proteins are ordered by explained variance across all factors. Colours indicate domains of explanatory factors. (D) Uniform manifold approximation and projection (UMAP) mapping of the variance explained matrix across 2853 protein targets for which we identified at least one feature explaining the variance in plasma levels. Each protein has been assigned a cluster based on k-means clustering and is coloured accordingly. (E) Violin plots showing the explained variance across protein targets according to categories of selected characteristics. The centre of each violin plot is a boxplot giving the median (white dot) and interquartile range. The boxplots were drawn using default options: lower whiskers = 25th percentile − 1.5 x interquartile range; upper whiskers = 75th percentile + 1.5 x interquartile range; centre = 50th percentile (median); lower box bound = 25th percentile; upper box bound = 75th percentile; minima and maxima represent the most extreme values and are plotted as outliers if exceeding whiskers. Explained variance distribution for a total of 2919 protein targets are displayed. Data information: In (E), data were presented as boxplots, indicating median, interquartile range (IQR), and whiskers for >1.5 times IQR. Source data are available online for this figure.

specific biomarkers and drug response markers from observational studies. We create a resource that is publicly available to the community to explore results (https://omicscience.org/apps/prot_foundation/) and a statistical framework to implement phenotype enrichment analyses in external studies (https://github.com/comp-med/r-prodente).

# Results

We identified 411 diverse, modifiable and non-modifiable, participant and technical characteristics (out of a total of 1879; Fig. 1B; Dataset EV1) to explain variation in plasma levels of one or more of 2853 protein targets (97.7%; Dataset EV2) among 30,268 UK Biobank participants (54.1% female; 5.2% non-European) using regularised linear regression models with stability selection (Meinshausen and Bühlmann, 2009). A median of 20 characteristics (range: 1–37) was selected across these protein targets and cumulatively explained on average 19.8% (range: 0.0005–100.0%) of variation in plasma protein levels in an independent validation set ($n = 12,972$; Fig. 1C; Dataset EV3). The highest amount of variance was explained for proteins actively secreted into blood (median of 25.5%) and those reliably detected using the assay technology (median of 20.2% for 1990 proteins with ≤5% of values below the limit of detection). The ability to reliably detect mRNA levels of the protein-coding gene in one or more tissues was further associated with a higher amount of explained variance. (Fig. EV1A–H).

We sought replication of our results in our previous work in a different cohort (Carrasco-Zanini et al, 2024b). Despite differences in the applied proteomic technology and available participant characteristics, the total amount of explained variance was significantly correlated ($r = 0.28$; $p$ value $\leq 6.7 \times 10^{-37}$) across 1968 protein targets measured on both platforms (Dataset EV4). The correlation coefficient improved to 0.46 ($p$ value $\leq 2.8 \times 10^{-37}$) among 699 protein targets that have been reported to correlate well across proteomic technologies (Eldjarn et al, 2023), demonstrating good generalisability of our results. Improvements in explained variance by the present study were most strongly associated with the inclusion of disease status (+14.4% per 1%-increase in explained variance by disease status; $p$ value $\leq 4.8 \times 10^{-17}$), genetically inferred ancestry (+8.5%; $1.3 \times 10^{-31}$), or considering parameters of general health (+18.4%; $p$ value $\leq 9.8 \times 10^{-7}$). Results that demonstrated the importance of expanding the phenotypic space compared to our pioneering work.

## A coordinate system for plasma protein variation

Projecting the entire matrix of proteins times explaining factors into a lower-dimensional space established a coordinate system along which proteins segregated into eight distinct clusters that we labelled according to shared major influences but also tissue and cell-type origin (Fig. 1D). The largest cluster showed evidence for enrichment by multiple explanatory factors, whereas remaining clusters distinguished by at most a few or even single characteristics explaining most of the variance in plasma levels of proteins within the cluster (Dataset EV5; Fig. EV2). For example, one cluster contained 617 protein targets that were strongly enriched for participant and sample characteristics, indicating effects of technical variation and contamination by blood cell activation.

For example, recruitment centre (beta: 0.18; $p$ value $<4.9 \times 10^{-08}$; mean explained variance (MEV): 1.82%) or plateletcrit (PCT) (beta: 1.73; $p$ value $<2.8 \times 10^{-264}$; MEV: 7.2%) were most commonly selected for those proteins (Fig. EV2K,L). Our findings are in line with previous studies (Geyer et al, 2019; Yunga et al, 2022; Korff et al, 2025) and are likely a result of platelet activation, and the subsequent release of contained proteins into plasma (Yunga et al, 2022). Collectively, these results provide evidence, that protein signatures in blood can be partly deconvoluted into distinct origins. Modifiable participant characteristics thereby outweighted non-modifiable ones, such as age, genetic sex, ancestry, or common protein quantitative trait loci, on average ($p < 7.5 \times 10^{-47}$; two-sided Wilcoxon rank-sum test; Fig. 1E).

## Organ and cell-type contributions

Proteins circulating in blood have diverse origins, and we therefore integrated gene expression data (Uhlén et al, 2015; Karlsson et al, 2021) to understand whether the selection of participant characteristics can be explained by effects in certain tissues or cell types. More than 80% ($n = 2420$) of all detected protein targets showed evidence that the corresponding mRNA is preferentially expressed in one or at most a few tissues or cell types, enabling enrichment analysis in up to 13 tissues and 28 cell types (Dataset EV2). This identified 94 characteristics significantly more frequently selected for proteins with tissue or cell type-specific mRNA expression (Figs. 2 and EV3; Dataset EV6, 7). Many of these enrichments were likely driven by tissue or cell damage. For example, proteins with enhanced mRNA expression in the liver, specifically hepatocytes, were explained by medications with adverse hepatic effects such as oral contraceptives (Kalman, 1969) (e.g. conjugated oestrogens; odds ratio: 5.62; $p$ value $<3.3 \times 10^{-18}$) or carbamazepine (odds ratio: 8.98; $p$ value $<1.3 \times 10^{-13}$). Other medication associations more likely represented effects on target organs/cells, such as proton pump inhibitors and stomach (e.g. esomeprazole; odds ratio: 237.10; $p$ value $<3.9 \times 10^{-15}$) or immune suppressants and immune cells (e.g. azathioprine; odds ratio: 13.01; $p$ value $<2.3 \times 10^{-16}$).

On a cell population level, we observed yet unreported links between participant characteristics and proteins with enhanced mRNA expression in fibroblasts that synthesise the extra cellular matrix (ECM; Fig. 2). This included drug usage like beclomethasone dipropionate (odds ratio: 14.24; $p$ value $<4.0 \times 10^{-9}$), diseases like atrial fibrillation (odds ratio: 6.13; $p$ value $<2.7 \times 10^{-9}$), but also participant's age (odds ratio: 2.88; $p$ value $<6.7 \times 10^{-10}$), body mass index (BMI; odds ratio: 3.49; $p$ value $<7.7 \times 10^{-13}$) and genetically inferred ancestry (odds ratio: 2.71; $p$ value $<7.2 \times 10^{-9}$). These results provide evidence that changes in plasma protein levels can, to some extent, be explained because of adverse, but also intentional, effects of drugs and diseases on specific tissues or cell types.

## Sex- and ancestral differential effects on the plasma proteome

Genetically inferred ancestry ($n = 1139$) and genetically inferred sex ($n = 1199$) were among the most frequently selected non-modifiable participant characteristics. To better understand potentially differential or specific effects across ancestries (41,369 White Europeans, 849 British Africans and 823 British Central South Asians) and the

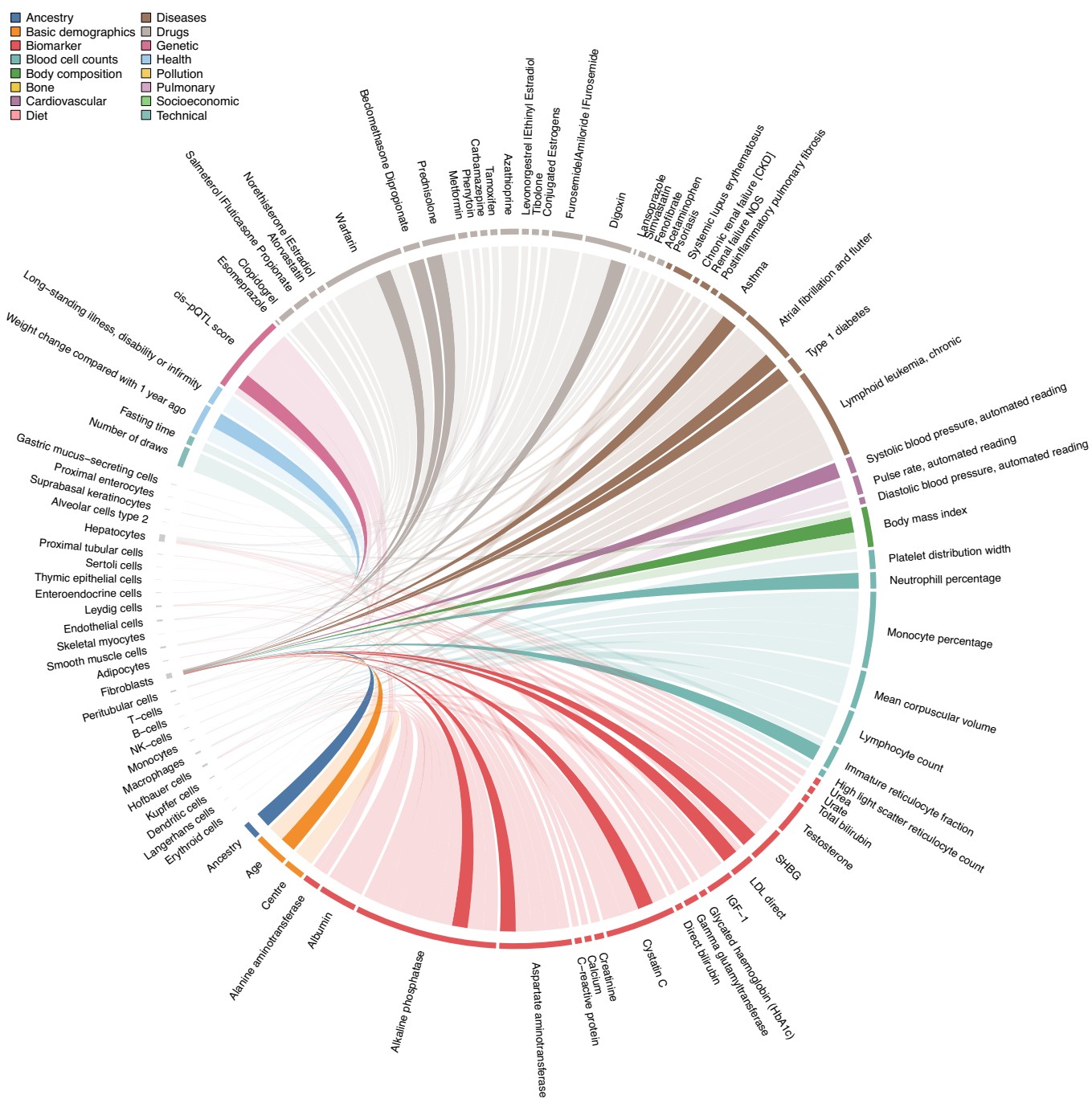

**Figure 2. Plasma proteins link cell types to indicators of health and disease.**

Chord diagram of phenotype-associated protein signature enrichment among protein-coding genes with enhanced cell type expression ('marker genes'). Each line represents a significant enrichment (Fisher's test; $p < 2.9 \times 10^{-6}$) of proteins associated with a participant characteristic among protein-coding genes with enhanced expression in a cell type. Enhanced expression estimates were derived from single-cell RNA sequencing data in the Human Protein Atlas. Corresponding statistics can be found in Dataset EV7. Associations with marker genes of fibroblasts have been highlighted by darker colours. Source data are available online for this figure.

sexes (23,601 females and 20,055 males), we repeated the feature selection procedure separately within each of the groups.

We identified significantly lower levels of explained variance in participants of British African (median: −5.07%; IQR: −12.64 to −0.03%; $p < 1.5 \times 10^{-91}$) and to a much lesser extent British Central South Asian ancestry (median: −0.03%; IQR: −4.36%–3.90%;

$p < 1.3 \times 10^{-2}$) compared to White Europeans. Results, that were, at least among participants of British African ancestry, not entirely explained by the lower sample size. We still observed a difference of ≥1.4% in explained variance in plasma levels for more than half of the protein targets following matching for the number of selected features across the White-European and British African cohorts

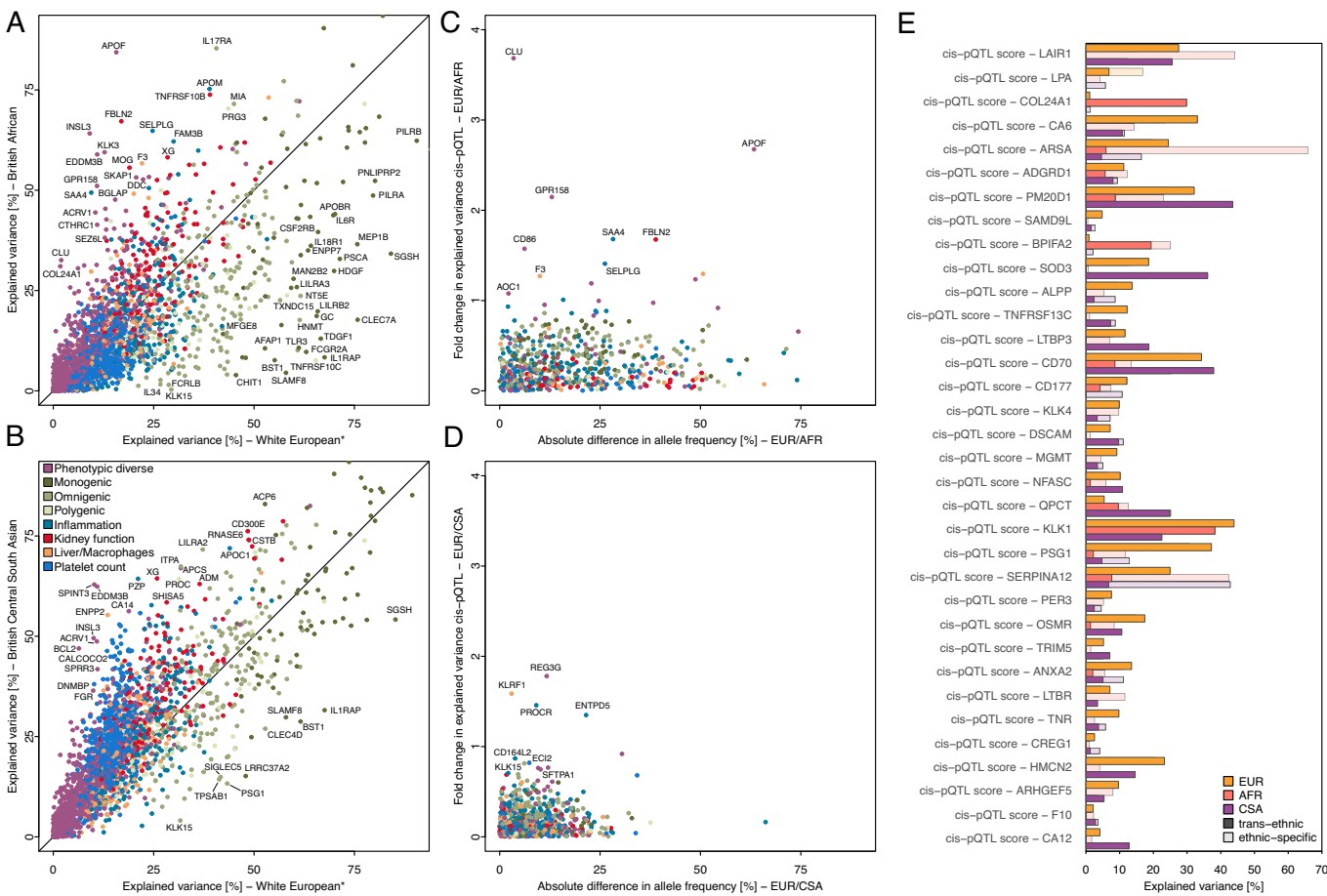

**Figure 3. Summary of ancestral-stratified analysis.**

(A) Comparison of the achieved explained variance in plasma proteins levels within White Europeans compared to British African participants. Proteins are coloured based on cluster membership, and those with extreme differences are annotated. *The number of selected features has been matched with the smaller ancestry to account for differences in the power of discovery. (B) Same as (A), but now comparing to participants of British Central South Asian ancestry. (C, D) Contrasting differences in allele frequencies for cis-pQTL scores with changes in explained variance between White Europeans and British African (C, AFR) or British Central South Asian (D, CSA). (E) Variance explained estimates for 34 cis-pQTLs with evidence for distinct ancestral-lead signals. Darker colours indicate the effect of the trans-ancestral signal, whereas shades indicate the explained variance by the ancestry-specific lead signal. We note that the effect of ancestry-specific lead signals might be overestimated, since they have not been selected from an independent cohort. Data information: In (A, B), point estimates were derived as variance explained ($r^2$) from a multivariable linear regression model; In (E), partial variance explained was computed for each genetic variant based on a multivariable linear regression model. Source data are available online for this figure.

(IQR: −6.58–3.15%, $p$ value $<1.0 \times 10^{-66}$; Fig. 3A,B) or when rerunning feature selection in age- and sex-matched subsets of Europeans with the same sample size (median: −2.62%; IQR: −9.34–0.00%; $p$ value $<6.1 \times 10^{-29}$).

The single strongest contributor of differential estimates of explained variance in plasma protein levels across ancestries was the contribution of cis- and trans-pQTL scores (Dataset EV8). Possible explanations include that pQTLs are present at different minor allele frequencies (MAF; Fig. 3C,D) across ancestries or may have differential effects on protein levels. We identified 34 protein targets for which frequency-enriched ancestry-specific lead signals (non-overlapping haplotype blocks; Fig. 3E) accounted for strong differences in explained variance. For example, the missense variant rs2071421 (p.Asn352Ser) conferring arylsulfatase A (ARSA) pseudo deficiency was most common in participants of British African ancestry ($MAF_{AFR} = 38.1\%$) but less frequent in participants of

other ancestries ($MAF_{EUR} = 11.0\%$; $MAF_{CSA} = 14.8\%$) and accordingly explained considerably more variance (65.8 vs 5.9%) compared to the trans-ancestral-lead signal rs873697 ($MAF_{AFR} = 11.5\%$; $MAF_{EUR} = 4.4\%$; $MAF_{CSA} = 0.6\%$) in participants of British African ancestry. However, most ancestral differential effects, including 135 out of 137 cis-pQTLs, were due to the same pQTL having different effect sizes across ancestries, with no other nearby variant explaining those. The reason for the different effects of the same genetic variant across ancestries remains to be established.

We lastly identified a few protein targets ($n = 22$) that were much better explained (>4 s.d.) in one but not the other genetically inferred sex (Fig. EV4A–C). We, however, noted that selected participant characteristics differed for almost a third ($n = 895$) of the protein targets (median Jaccard index = 0.56; Fig. EV4C), indicating the need to consider sex-specific contributions to plasma protein levels (Dataset EV9). Obvious examples included

medications ($n = 1062$ pairs, most frequently anticontraceptives) and diseases ($n = 30$ pairs) given/occurring only in one sex, whereas abundant sex-differential effects were explained by associations with age, biomarkers, or body mass index (Fig. EV4D; Dataset EV9). Notably, there were only a few examples of sex-differential genetic effects, e.g. the cis-pQTL score for plasma oxytocin (OXT; female: 18.7%; male: 30.5%; $p$ value$_{inter}$ $<2.8 \times 10^{-48}$; Fig. EV4E).

## Protein biomarker discovery and pruning for incident diseases

We next systematically explored how our results can guide the identification of plasma protein biomarkers in biobank-scale studies. Among the 67,033 significant protein–disease associations ($p < 4.1 \times 10^{-8}$) observed here and reported in other studies (Eldjarn et al, 2023; Deng et al, 2024), we observed a more than 32-fold drop after regressing out characteristics explaining variation in plasma protein levels for 424 incident diseases (Fig. 4A; Dataset EV10). Associations of more than two-thirds (1333 out of 2080) of protein targets were almost completely attenuated. Even among the 1975 protein–disease associations with directionally concordant effects and persisting significance, >80% ($n = 1691$) showed considerable attenuation of effect sizes in Cox models (≥20%). This suggests that more precise measurements of associated participant characteristics are likely to lead to further statistical attenuation and indicate that most protein–disease associations can be explained by common participant characteristics (Fig. 4B). Notably, a >5-fold decrease in the number of significant protein–disease associations was achieved with as few as five selected characteristics (67,033 to 13,307), demonstrating the importance of understanding and considering protein determinants rather than solely statistical significance for distinguishing biologically relevant from false positive findings and type 1 errors.

Robust protein–disease associations included established clinical screening markers such as prostate-specific antigen (referred to by Olink as KLK3) for prostate cancer (HR 3.11; $p$ value $5.3 \times 10^{-254}$), and markers of early tissue damage, such as the lung-specific surfactant protein D (SFTPD) for post-inflammatory pulmonary fibrosis (Ikeda et al, 2017), or the eye-specific protein crystallin beta B2 (CRYBB2) for cataract (HR 1.41, $p$ value $3.8 \times 10^{-97}$). Less-established links with strong effect sizes even after accounting for all selected phenotypic characteristics included a 2.3-fold increased risk ($p$ value $1.1 \times 10^{-38}$) for abdominal aortic aneurysm (AAA) per 1 s.d. increase in plasma levels of matrix metalloproteinase 12 (MMP12) (Figs. 4B and EV5). A putative role of MMP12 in the progressive degradation of the ECM at the aortic wall, a hallmark of AAA, was thereby supported by multiple lines of evidence. Firstly, we obtained evidence that the same genetic variant (rs17368814) that increases plasma MMP12 levels by acting on its protein-coding gene also increased the risk for AAA (Posterior probability shared genetic signal: 97.5%; Mendelian randomisation estimate: hazard ratio = 1.15; $p$ value = $3.3 \times 10^{-16}$; Fig. EV6), providing support that changes in plasma MMP12 precede AAA onset. Secondly, MMP12 was elevated among patients diagnosed with AAA even after adjustment for all factors explaining MMP12 levels (beta = 1.09 s.d. units, $p$ value $<2.1 \times 10^{-21}$), potentially indicating ongoing degradation of the ECM. Thirdly, our observational findings are in line with experimental evidence linking matrix metalloproteinases to the onset and progression of AAA (Hellenthal et al, 2009), with MMP12 being highly expressed at disease sites and suggested to contribute to the degradation of elastic fibres and hence weakening and dilation of the aortic wall (Curci et al, 1998). These findings seem to be context-specific, since pharmacological inhibition of MMP12 protected $Apoe^{-/-}$ deficient mice from AAA (Di Gregoli et al, 2024), while, paradoxically, $Mmp12^{-/-}/Apoe^{-/-}$ mice were more susceptible to AAA and subsequent rupture (Salarian et al, 2023).

Identification of putative protein biomarkers has previously been proposed using genetic imputation (Xu et al, 2023), suggesting that measuring common genomic variation can proxy measurement of plasma protein levels and hence guide tailored biomarker assessment. However, even scaled to the power of the entire UK Biobank cohort, we observed little overlap between associations with genetically imputed and measured plasma protein levels, including those that we robustly linked to disease outcomes (Fig. 4C,D; Dataset EV10). For example, the five most significant genetically proxied protein–disease associations were not among the twenty most strongly associated measured protein–disease associations for two-thirds of all diseases considered (268 out of 419). This included not only strongly differing effect estimates, and hence power to identify people at high risk early, but also discordant results whether considering cis- or trans-pQTLs for imputing plasma protein levels (Fig. 4C,D).

## A protein foundation community resource

Most plasma proteomic studies are done at a small scale, with often incomplete metadata on participants, adding to the complexity of the unknown sources of variation of most plasma proteins. We therefore developed a phenotype enrichment framework that allows us to test for significant enrichment (correcting for multiple testing using the Bonferroni procedure) of participant characteristics otherwise hidden in differentially expressed plasma proteomic signatures (https://github.com/comp-med/r-prodente).

As a proof of principle, we observed proteins associated with the UKB characteristic 'fasting time' to be more than 90-fold enriched (odds ratio: 91.0; $p$ value $<1.6 \times 10^{-5}$; Fig. 5A) among proteins significantly different following one day of complete caloric restriction in a well-controlled intervention study (Pietzner et al, 2024). However, a similar enrichment of a protein signature linked to plasma bilirubin was not reported due to missing measurements but resembles the well-known increased reuptake of bilirubin via the enterohepatic cycle due to lower gut motility during prolonged fasting (Gambino, 1972; Barrett, 1971).

Another important application of phenotype enrichment is the discovery of unknown confounders or imperfect matching in biomarker studies. For example, we observed a more than twofold enrichment of proteins associated with smoking among those differentially expressed between patients with ovarian carcinoma and selected controls (Qian et al, 2024) (Fig. 5A). Accordingly, smoking status was among the top three variables explaining plasma variation in the most differentially expressed proteins in patients with cancer, Kunitz-type protease inhibitor 1 (SPINT1) (3.0%; Fig. 5B). We observed similar residual phenotypic enrichments, including smoking or socioeconomic factors, in a plasma proteomic model to predict future coronary artery disease (Helgason et al, 2023) (Fig. 5B), or among recently proposed proteomic organ clocks (Oh et al, 2023). Those proxying pancreatic

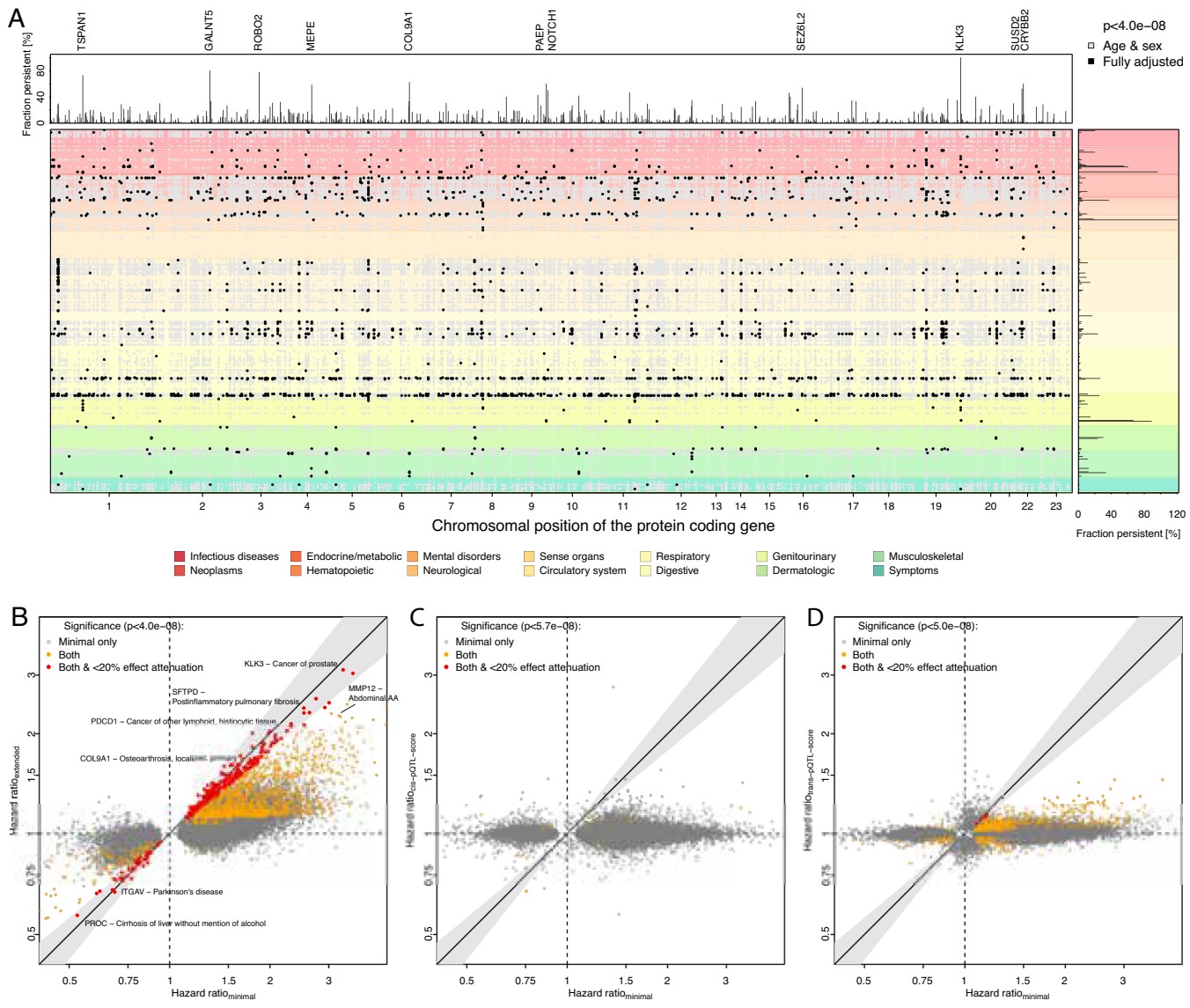

**Figure 4. Protein–incident disease associations.**

(A) Summary of associations between plasma protein levels (x-axis indicated by the position of the protein-coding gene) and one or more of 424 diseases in UK Biobank using Cox-regression models passing multiple testing corrected statistical significance (Cox-proportional hazard model; $p < 4.1 \times 10^{-8}$). Each dot represents an association passing multiple testing, and black dots indicate those persisting after regressing out factors that explained plasma protein variation. The top panel illustrates the fraction of disease associations per protein that were still significant after accounting for associated protein characteristics, similar to the right panel for diseases. Proteins or diseases with minimal effect attenuation were annotated. (B) Scatterplot comparing hazard ratios per 1 s.d. increase in protein levels from Cox-regression models adjusting for age and sex, or additionally accounting for factors explaining variation in plasma protein levels. Only protein–disease associations passing multiple testing corrected statistical significance in at least the minimal model are shown and coloured according to significance and effect attenuation in the extended model. (C, D) Scatter plots opposing effect estimates for significant (see legend for $p$ value threshold) protein–disease associations comparing those using measured plasma protein levels (x-axis; $n = 43,647$) to those based on genetically imputed plasma protein levels in the entire unrelated White-European UKB cohort (y-axis; $n = 342,240$). Data information: In (A–D), each dot reflects a significant association between plasma protein levels and disease onset based on Cox-proportional hazard models. Exact $p$ values and association statistics can be found in Dataset EV10. Source data are available online for this figure.

or intestinal age being enriched for established markers of muscle mass (odds ratio: 28.3; $p$ value: $3.6 \times 10^{-10}$) or fasting time (odds ratio: 28.2; $p$ value $<1.6 \times 10^{-9}$), respectively (Fig. 5C).

Most notably, all studies showed strong depletion of proteins associated with technical characteristics, such as study centre, suggesting that protein biomarkers emerging from well-controlled settings are less likely to be driven by analytical artefacts. However, in study designs with separate blood sampling of patients and controls, such as in the blood disease atlas from the Human Protein Atlas project (Uhlén et al, 2015), we observed strong enrichment of such characteristics, dominating biologically plausible findings (Fig. EV7).

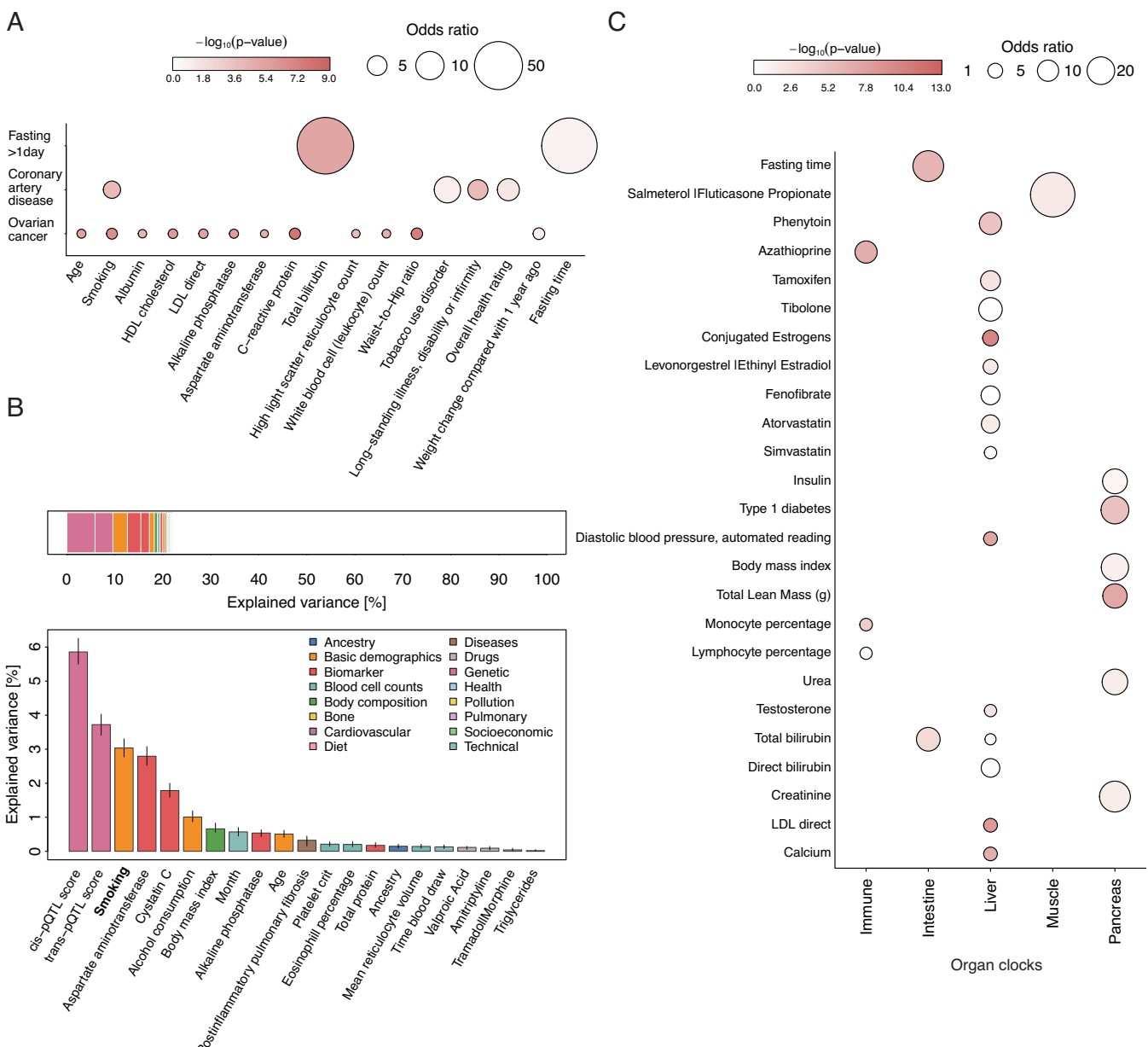

**Figure 5. Phenotype enrichment of plasma proteomic signatures.**

(A) Phenotype enrichment for plasma proteomic signatures that (1) differed after one day of complete caloric restriction, (2) differentiated healthy women from women with ovarian cancer (Qian et al, 2024) and (3) a proteome score to predict the onset of coronary artery disease CAD (Helgason et al, 2023). Enrichment was computed using Fisher's exact test, and only factors passing corrected statistical significance are shown (Bonferroni correction). (B) Factors explaining variance in plasma levels of serine peptidase inhibitor, Kunitz-type 1 (SPINT1), one of the most differential plasma proteins described for ovarian cancer. Values were derived from 23,067 female UKB participants. Error bars represent the 2.5th and 97.5th percentiles from 200 bootstrapping iterations to compute the explained variance. (C) Phenotype enrichment among plasma proteomic signatures proposed to track organ age (Oh et al, 2023). Data information: In (A, C), enrichment statistics were derived based on Fisher's exact test; In (B), partial variance estimates were derived from a multivariable linear regression model. Source data are available online for this figure.

## A knowledge graph to triangulate evidence for clinical impact

We finally integrated results from the multivariable variance estimation with drug target annotations, pQTLs and potential effector genes, along with genetic disease associations within a shallow knowledge graph to visually and dynamically illustrate examples of potential clinical impact (Fig. 6A; Datasets EV11, 12). This included 34 examples in which plasma protein levels might act as readouts of successful drug target engagement (Fig. 6B). We identified those as subnetworks that link high-confidence drug–protein associations from the present study with genetic variants that mimic drug target modulation by changing the expression or function of the drug target and are associated with the same

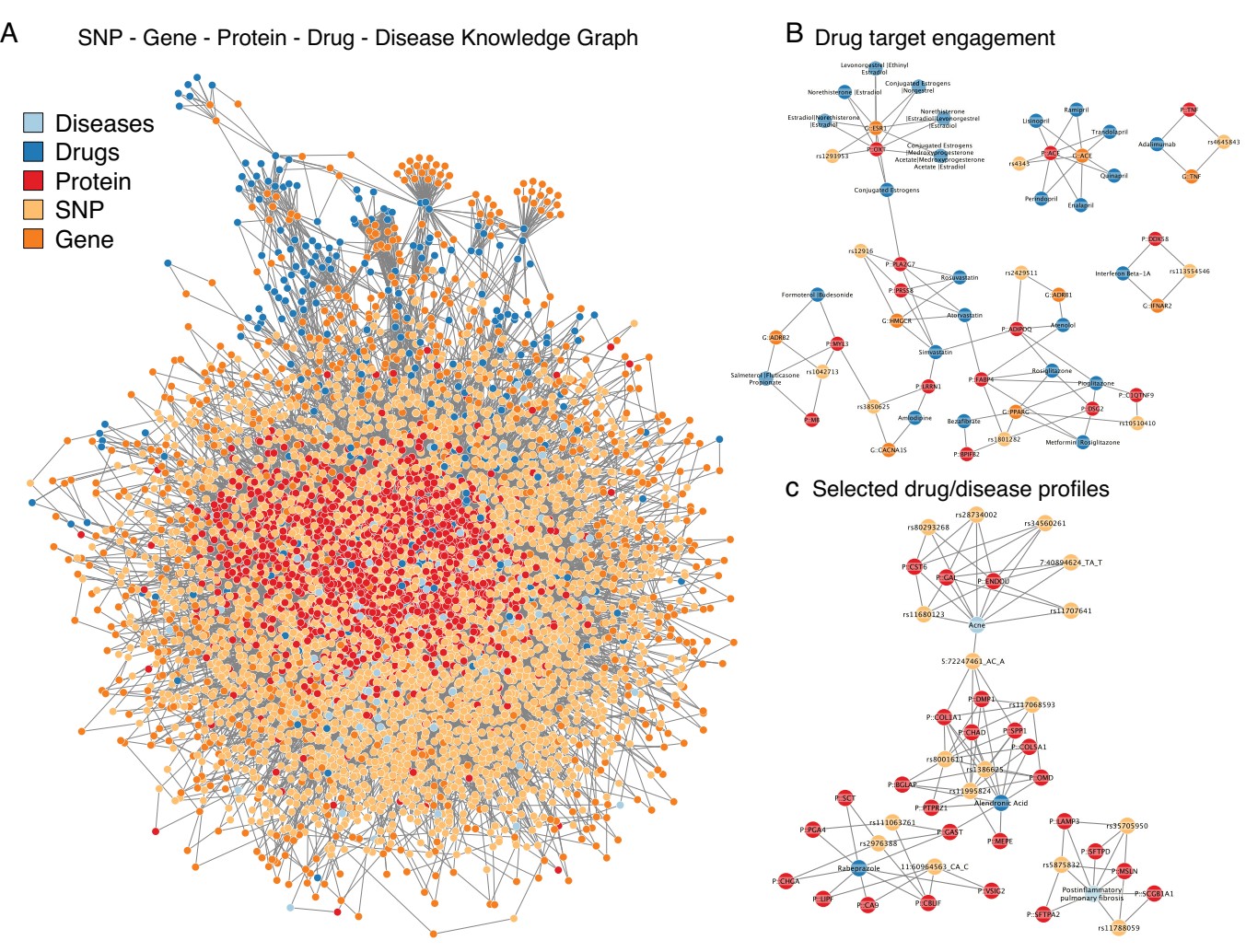

Interactive version https://omicsscience.org/apps/prot_foundation/

**Figure 6.  Integrated knowledge graph of gene, protein, drug and disease relationships.**

(A) Full knowledge graph connecting single-nucleotide polymorphisms (SNPs – pQTLs) to proteins and diseases from previous publications (Sun et al, 2023) and the GWAS catalogue (Buniello et al, 2019). Each SNP association represents a genome-wide significant finding. Drugs were mapped to target genes based on Open Targets (Ochoa et al, 2023). Protein–disease and protein–drug associations were derived from the present study. The graph was visualised using Cytoscape v.3.10.3. (B, C) Specific subnetworks derived from the knowledge graph illustrating evidence for drug target engagement markers (B) by linking pQTLs to drug target genes, and non-random pQTL —drug protein profiles (C). An interactive version of (A) can be found at https://omicsscience.org/apps/prot_foundation/. Source data are available online for this figure.

protein(s) (Fig. 6B). For example, plasma levels of desmoglein 2 were explained by pioglitazone intake (0.2%, beta = 1.02 s.d. units, $p$ value = $3.7 \times 10^{-39}$) and the corresponding pQTL, rs1801282 (beta = 0.06; $p$ value $<1.4 \times 10^{-12}$), maps to *PPARG* encoding the pioglitazone target peroxisome proliferator-activated receptor gamma. Other such examples included plasma oxytocin as a readout for oral contraceptives or adiponectin as a readout for atenolol treatment. The latter had previously been suggested from small-scale trials in a specific target population (Pöyhönen-Alho et al, 2008), whereas a role of desmoglein 2 in the regulation of beta-cell activity has been proposed only recently (Myo Min et al, 2022).

In general, enrichment of proteins differentially expressed in disease states for associations with germline genetic variants, i.e.

pQTLs, can point to causal relationships. We observed 595 protein - disease/medication examples with significant support from enrichment analysis (Datasets EV13, 14). For example, cystatin E/M (CST6), endonuclease, poly(U) specific (ENDOU), and galanin (GAL) were associated with acne in our study (e.g. galanin: beta = 0.27 s.d. units, $p$ value $<1.2 \times 10^{-13}$) and six independent pQTLs, which themselves have been reported to increase the risk for acne (Fig. 6C; Dataset EV13). The robust link between acne and the neuropeptide galanin aligns with its expression in non-neural tissues such as skin (Kofler et al, 2004), where it can modulate inflammation in a context-specific manner (Lang et al, 2015). Although no link to acne has been made so far. Similar triangulation with medication intake provided further strong support for the role of platelet modulation on the

plasma proteome, with 41 genetic loci being significantly enriched for proteins associated with the intake of the antiplatelet medication clopidogrel, almost all of which have been previously linked to platelet characteristics (Dataset EV14). We observed similar convergence of protein profiles for genetic predisposition to and effects of drugs prescribed for osteoporosis (e.g. bisphosphonates like alendronic acid linked to five independent pQTLs) or gastric ulcers (e.g. proton pump inhibitors like rabeprazole linked to three independent pQTLs) (Fig. 6C), providing new avenues to understand drug effects. An interactive version of the knowledge graph can be explored in a customised fashion to identify other such examples (https://omicscience.org/apps/prot_foundation/).

## Discussion

Deep phenotyping of biobank-sized cohorts now provides unprecedented opportunities to identify novel protein biomarkers for common and rare diseases (Carrasco-Zanini et al, 2024a) and to inform drug target discovery (Sun et al, 2023). However, emerging production-type protein biomarker association analyses are already repeating failures of decades of biomarker research, as they rarely translate into actionable knowledge. We created a data-driven model of plasma protein variation that revealed cell type-specific, but also systemic effects and provides a foundation to understand how variation in blood protein levels is linked to human health. We demonstrated its importance in several ways, including biomarker identification for the onset of severe diseases, drug target engagement markers, and as a community resource to complement pathway enrichment analysis, providing otherwise unattainable insights. We created a community resource to enable interactive exploration (https://omicscience.org/apps/prot_foundation/) and incorporation in plasma proteomic workflows (https://github.com/comp-med/r-prodente).

Although not directly comparable, we prioritised ten times less protein–phenotype combinations (~50,000 vs >500,000) compared to similar efforts within UKB (Eldjarn et al, 2023; Deng et al, 2024) emphasising the need to consider and quantify the effect of different factors simultaneously rather than in isolation to overcome confounding. Only such an increase in the specificity of phenotype–protein associations enabled tangible phenotype enrichment analysis, and subsequently the identification of disease-specific biomarkers that are not only statistically significant, but relevant. The same applies to the considerable amount of sharedness of protein–disease associations proposed previously (Deng et al, 2024), which we demonstrated to be largely driven by known phenotypic characteristics. Notably, phenotype enrichment analysis of sparse predictive biomarker signatures previously derived by our group (Carrasco-Zanini et al, 2024a) showed no evidence for widespread enrichment of phenotypic characteristics. It rather pointed to tissue damage markers or refinement of imprecise phenotypic risk factors such as smoking or socio-economic status (Dataset EV15). At the same time, we provided evidence that purely genetically-anchored strategies (Xu et al, 2023) are unlikely to discover powerful protein biomarkers as they lack the dynamic component.

Including >1800 participants and technical characteristics explained on average about a quarter of the variance in plasma protein levels across the population. Based on our findings, we hypothesise that factors we had not considered or were not assessed in the UK Biobank will likely substantially increase the explained variance for at most few targets. Significant improvements in the explained variance across all protein targets are more likely to be achieved by improving assay performance and sample handling, but also by measuring exposures more precisely. For example, quantifying the individual exposure to environmental pollutants instead of regional measures or quantifying drug exposure using dosages instead of self-report.

While we were underpowered to quantify ancestral-differential effects at scale, the proteins we identified were either explained by varying effect allele frequencies or differential effect sizes of the same genetic variant across ancestries. The former explains ancestral-enriched or even specific phenotypic consequences, such as for beta-thalassaemia, whereas the latter likely indicates the presence of yet to be identified gene–environment interactions. Larger studies are needed to establish the transferability of our results to different population subgroups to ensure equitable biomarker research.

Pathway enrichment has been transformative in genome-wide differential gene expression studies to enable interpretation of otherwise massive gene lists and has been almost seamlessly adopted for proteomic studies. While such analyses can guide tissue or single-cell studies, proteins found in plasma are of diverse origin, making the interpretation of such findings difficult and possibly even misleading. We demonstrated across different applications, how our results can guide the interpretation of plasma protein biomarker studies, including novel insights but also pitfalls when differences in sample handling coincide with case–control comparison. Future extensions may incorporate different proteomic technologies and biofluids to enable widespread application.

The advances of our study have to be considered according to a number of limitations. Firstly, our results remain restricted to the selection and precision of measurements of participant and sample characteristics. Other data modalities, such as imaging of organs or exposure to different environments or controlled challenge studies, will likely reveal additional contributions to the plasma proteome. Secondly, we cannot entirely rule out that selected participant characteristics may only approximate truly underlying reasons for variation in plasma protein levels, and that observed associations, even if pertaining to statistical significance in multivariable linear regression models, do not necessarily represent causal associations. Thirdly, we observed evidence that variation in plasma levels of proteins close to the limit of detection were less well explained, suggesting that more sensitive techniques for such protein targets may improve estimations of explained variance. Lastly, while cis-pQTLs provide reassurance in assay specificity, we cannot completely rule out that comparatively high amounts of variance explained by such cis-pQTLs might be the result of measurement artefacts. This may imply that variance estimates for such targets therefore relate to a specific isoform of the protein target. However, our approach to mutually model genetic and non-genetic effects delivered reliable estimates for the latter, even if the genetic part might have been inflated.

# Methods

**Reagents and tools table**

| Reagent/ resource | Reference or source | Identifier or catalogue number |
|---|---|---|
| **Experimental models** | | |
| N/A | | |
| **Recombinant DNA** | | |
| N/A | | |
| **Antibodies** | | |
| N/A | | |
| **Oligonucleotides and other sequence-based reagents** | | |
| N/A | | |
| **Chemicals, enzymes, and other reagents** | | |
| N/A | | |
| **Software** | | |
| R v.4.3.0 | https://cran.r-project.org/ | |
| Python v.3.1 | https://www.python.org/ | |
| Bgenix v.1.0 | https://enkre.net/cgi-bin/code/bgen/ doc/trunk/doc/wiki/bgenix.md | |
| Qctool v2.0.7 | https://www.chg.ox.ac.uk/~gav/qctool_v2/ documentation/changes.html | |
| Cytoscape v3.10.3 | https://cytoscape.org/ | |
| **Other** | | |
| N/A | | |

## Study participants

UKB is a large-scale, population-based cohort with deep genetic and phenotypic data, with the full cohort consisting of ~500,000 participants aged 40 to 69 years at the time of recruitment and has been described in detail elsewhere (Sudlow et al, 2015). We used the subset of individuals from UKB where both genotype and proteomic measurements were available after excluding ancestry outliers or samples which have failed genomic or proteomics quality control (*n* = 43,240). This research has been conducted using the UK Biobank Resource under Application Number 44448.

## Proteomic measurements

We used plasma proteomic measurements provided by UKB as described in detail elsewhere (Sun et al, 2023). After initial quality control checks (e.g. principal component analysis), we only retained individuals with at least 50% valid protein measurements (*n* = 43,240). Notably, UKB provided proteomic data with values flagged as assay warnings by Olink as 'NA', which meant that each protein measurement had differing numbers of valid values, and we decided not to impute values for the purpose of this study to minimise skewed estimates of features. We included a total of 2919 unique protein targets in our analysis.

## Phenotyping

We collated a large set of phenotypic and sample characteristics available for at least half of the UKB participants (Dataset EV1). Those included information on genetically inferred ancestry (see below), basic demographics (e.g. age and sex, *n* = 5), blood-based biomarkers (e.g. low-density lipoprotein cholesterol, *n* = 28), blood cell counts (*n* = 29), body composition (*n* = 20), indicators of bone health (*n* = 7), cardiovascular risk factors (*n* = 3), diet (*n* = 23), pre-existing diseases (*n* = 1198), self-reported medication intake (*n* = 704), generic indicators of health (*n* = 4), indicators of pulmonary health (*n* = 8), information on air pollution (*n* = 10), socioeconomic indicators (*n* = 9), and technical variables (*n* = 5), following pruning as described below.

Measurement of blood-based biomarkers had been described in detail previously (Sinnott-Armstrong et al, 2021), and we further implemented standard quality control procedures to avoid strong influences of single values. We removed values more than five times the median absolute deviation away from the respective sample median (median of ≤0.05% values removed). Biomarkers with strong indications of skewed distributions were log-normalised (similar to protein values). We applied a similar quality control workflow to blood cell counts. We obtained information on when the blood sample was taken during the day, the age of the sample when proteomics measurements were done, and the duration since the last meal ('fasting time') as technical variables.

We compiled additional measures of body composition by augmenting Dual-X-ray predictions of lean and fat mass using equations provided by Powell et al (Powell et al, 2020) for each sex separately. In addition, we computed the waist-to-hip ratio as a common, readily available measure of body composition.

To identify participants with pre-existing diseases, we collated information from ICD-coded self-reports, hospital episode statistics, cancer registry, and primary care (for 45% of the population). We parsed all records to exclude codes with a recorded date before or within the year of birth of the participant to minimise coding errors from electronic health records. We mapped different coding systems to a total of 1198 'phecodes' that represent medical ontology terms intended to reduce redundancy among ICD-10 coding systems (Bastarache, 2021). We recorded the earliest occurrence of each 'phecode', referred to hereafter as 'disease' for simplicity, and created binary variables indicating whether it had been recorded before the baseline examination of the volunteer.

We obtained mappings of self-reported brand names to ATC codes from previous work to generate corresponding medication variables (Wu et al, 2019). We kept a total of 704 unique ATC codes mapping to medications taken by ten or more participants.

If not otherwise indicated, information was derived from basic questionnaires or the data portal (e.g. dietary habits and blood pressure readings) with minimal cleaning. We obtained regional lead variants explaining variation in plasma protein levels from the trans-ancestral meta-analysis performed by Sun et al (Sun et al, 2023) to compute cis- and trans-pQTL scores, which also served as genetically imputed protein levels. We used the regression estimates provided to compute a weighted genetic risk score for each individual for each protein with at least one reported pQTL.

After initial data collation, we filtered for variables with a correlation coefficient <0.85, retaining among highly correlated

variables the one with the least missing values or manually curated to maintain interpretability. This left us with a total of 1876 variables. Since feature selection required a complete data matrix, we imputed missing values in phenotypic data using random forest models that can cope with a mixture of different data types better than usual linear regression techniques (Stekhoven and Bühlmann, 2011). We used all selected phenotypic features for imputation, even if they had no missing values, to ensure the best predictive models for imputation. We further imputed the entire data set before splitting into sets for feature selection and variance estimation to ensure coherent estimates. We implemented these using the R package miceRanger (v.1.5.0). To ensure reasonable computation times, we performed imputation only once.

## Genotyping and ancestral assignment

We obtained genetic variants from imputed genotype information as described in detail elsewhere (Bycroft et al, 2018). We used the ancestry assignments as defined by the pan-UKB (Karczewski et al, 2024), and further assigned unclassified individuals to their respective ancestries based on a k-nearest neighbour approach using genetic principal components.

## Statistical analysis

To select approximately independent participant and sample characteristics that may collectively best explain variation in plasma protein levels, we implemented stability selection (Meinshausen and Bühlmann, 2009) and subsequent variance partition (Friendly, 2007). We first split UKB into a set for feature selection (70%) and used the R package stabs (v.0.6.4) to implement feature selection using regularised linear regression with the least absolute shrinkage and selection operator (LASSO). We controlled the upper bound per-family error rate at 1, used a cut-off of 0.75, and a maximum of 500 iterations. We controlled the output from stability selection by introducing 10 random variables, but observed consistent behaviour across the study (i.e. random variables were not selected as important). Based on the features selected, we next performed variance decomposition on 200 bootstrap samples of the remaining UKB set not used for feature selection (30%) and computed the partial explained variance using the *etasq()* function of the heplots R package (v.1.7.5) as recommended previously (Garcia et al, 2024). We finally used the median of variance estimates across all 200 bootstrap samples as an estimate for the explained variance in plasma protein levels and provide the 2.5th and 97.5th percentils as confidence intervals.

We repeated the same procedure using different strata of UKB (biological sexes and genetically inferred ancestry). To establish whether differentially selected features indeed corresponded to differential effects across strata, we additionally implemented interaction testing by performing linear regression analysis with the respective protein target as outcome and including the strata, the differentially selected feature, as well as an interaction effect between both as explanatory variables. We subsequently report strata-differential effects if the *p* value for the interaction term passed multiple testing correction. This test was only done for variables not specific to only one of the strata, e.g. no interaction analysis was done for the selection of oral contraceptives across the

genetically inferred sexes. We further created four different subsamples of the White-European cohort to match the sample sizes of British African and British Central South Asian participants, accounting for age and sex distributions, and repeated the entire feature selection. We took the median of the explained variance across all four subsamples as a summary measure.

To understand whether protein characteristics may have contributed to different levels of explained variance, we collated information for all protein targets from UniProt via the R package queryup (v.1.0.5) as well as the Human Protein Atlas (Fig. EV1). We binarized information of protein characteristics from UniProt to facilitate numeric analysis. We further computed summary measures of plasma protein level distributions based on untransformed NPX values. We then used Boruta feature selection (Kursa and Rudnicki, 2010) to identify protein characteristics significantly explaining variation in the levels of explained variance across protein targets based on the entire UKB cohort. We used standard parameters apart from setting the maximum number of iterations to 500, to ensure decisions on most characteristics.

We used uniform manifold approximation and projection (UMAP) to visualise any potential underlying structure in the plasma proteome according to factors contributing to the explained variation in plasma protein abundances. UMAP is a dimensionality reduction technique which enables better preservation of the data's global structure compared to similar methods such as t-SNE. We applied UMAP on the matrix of explained variation by all participants, environmental and technical characteristics on all protein targets. We used default values for most parameters used by the algorithm in the UMAP R package. Custom configuration was done for the following parameters as follows: random_state (seed for random number generator) = 10, metric = 'pearson', n_epochs = 1000, init = 'random'.

To establish clusters of protein targets, we implemented a k-nearest neighbour clustering based on the full variance explained matrix, but excluding features selected ≤10 times to reduce dimensional burden. We selected the number of clusters by visual inspection of the UMAP mapping and corroborating with marker analysis using hurdle models. The latter was implemented using the *zeroinfl()* function of the pscl R package (v.1.5.9), and we modelled the cluster assignment as exposure and the explained variance by phenotypic characteristics as count data to account for thresholding of the explained variance between 0 and 100 and an inflation of zeros (i.e. characteristics not being selected).

Researchers were not blinded to the outcome variables (protein levels) to enable downstream analysis.

## Ancestral differential variants

For each cis-pQTL with significant evidence for ancestral-differential effects, we obtained ancestral-specific GWAS results (Sun et al, 2023) and retained the strongest regional variant for further testing (±500kb). We declared an effect as ancestry-specific, if the regional lead variant within the ancestry was not included among a list of highly correlated genetic variants with the other ancestral regional lead variants ($r^2 > 0.1$), basically testing for overlapping haplotypes across ancestries. We subsequently tested each ancestral regional lead variant for effects in other ancestries and computed the explained variance. We further tested, whether

the association between the variant and the corresponding protein differed by ancestry (significant interaction effects). The latter was important to establish whether different amounts of explained variants across ancestries were due to different allele frequencies or differential effect sizes of similarly frequent alleles.

## Generation of a knowledge graph

We generated a shallow knowledge graph to facilitate triangulation of evidence for potential causal relationships. The graph contained a total of 30,068 edges and 6591 nodes. Nodes included protein targets, pQTLs from Sun et al (Sun et al, 2023), genes (either drug targets or effector genes for pQTLs), drugs (based on ATC codes in the present study), and diseases. We linked proteins to diseases (protein–disease) and drugs (protein–drug) based on evidence from the present study. We included edges linking pQTLs to proteins, and further linked pQTLs to diseases based on overlap with variants with genome-wide significance ($r^2 \geq 0.8$; $p < 5 \times 10^{-8}$) reported in the GWAS catalogue (download: 22/05/2025). We retained only entries from the GWAS catalogue that we could match to diseases defined in the UK Biobank by linking Experimental Factor Ontology (EFO) terms to ICD-10 codes using the EMBL-EBI API. We obtained information from Open Targets (Ochoa et al 2023) to map drugs to targets (genes) and restricted drugs to the once available in UK Biobank by mapping CHEMBL identifiers to ATC codes. We finally used effector gene assignments from Sun et al. (Sun et al, 2023) to link pQTLs to effector genes. We implemented a user-friendly interactive version of the knowledge using Python. More specifically, we implemented the knowledge graph using the NetworkX Python package (Hagberg et al, 2008) (v.3.3), and we enabled the knowledge graph to be interactively accessible using the Pyvis Python package (Perrone et al, 2020) (v.0.3.1). Notably, we customised the JavaScript generated by Pyvis to enhance the functionality of the interactive knowledge graph, allowing the users to obtain information about the neighbourhood of the selected node and the shortest cycle path associated with it, computed using the breadth-first search algorithm.

## Protein–incident disease analyses

We systematically investigated prospective associations of protein targets with 424 diseases with more than 200 incident cases during a follow-up time of 10 years. For each protein–disease pair, we employed separate Cox-proportional hazards models to model the associations between baseline plasma protein levels and the time-to-event, adjusting for age and sex of the participant. In case of sex-specific diseases, we used only the relevant subset of the population. We computed and compared Cox models with protein levels in two different forms, i.e. crude protein levels and residuals of proteins after regressing out identified explanatory factors. For each protein, we ran a multivariable linear model including all selected characteristics as explanatory variables and subsequently used the residuals (the fraction of variation in protein levels not explained) from this model as updated exposure in Cox models. We computed similar residual plasma protein levels accounting for the top 3, 5, 10 and 15 selected features. We entered the protein levels as continuous variables after performing rank inverse normal transformation for each predictor to ensure comparability of

results. Hence, we report hazard ratios along with standard errors per increment in one standard deviation for each protein. To account for potential reverse causality, we excluded cases that occurred during the first 6 months. We considered associations significant when $p < 4.0 \times 10^{-8} = 0.05/(2919 \text{ proteins} \times 424 \text{ diseases})$. We computed time-varying effects for each protein–disease association by computing Schoenfeld residuals, as well as by restricting follow-up time to 2, 5 and 10 years. We used the R package survival (version 3.7.0) for these analyses.

## Mapping to tissues and cell types

We programmatically downloaded tissue expression data from the Human Protein Atlas (HPA) for all Olink proteins in JSON format (on 20.03.2024). We performed a two-sided Fisher's exact test to determine whether any participant characteristic was selected ≥5 times across all proteins, whether there was an enrichment of associated protein targets for those showing specific expression of corresponding mRNA levels in certain tissues or cell types. We defined tissue- or cell-type-specific as 'enhanced', 'enriched' or 'group enriched' according to HPA classification. We further performed a two-sided Fisher's exact test to determine whether proteins predicted to be secreted according to the HPA annotation were significantly enriched among any of the clusters of proteins explained by the same biological influence.

## A phenotype enrichment framework

To enable phenotype enrichment, we first collated all results across all strata (i.e. all participants, the sexes, and the three ancestries) tested into one results file (Dataset EV3). For each stratum, we then implemented Fisher's exact test to test for an enrichment of a specified list of protein targets among protein signatures associated with participant and sample characteristics tested in our work. In other words, for a given list of proteins, we ask the question whether there are participant characteristics more frequently selected than by chance to explain variation in their plasma levels. We restricted enrichment tests to participant and sample characteristics with at least five associated, e.g. selected, protein targets. We used dot plots to visualise results. All analyses have been implemented in the R package prodente (https://github.com/comp-med/r-prodente) for dissemination. For plasma proteomic studies not using the Olink Explore 3072 platform, we used a specific background list of shared protein targets to minimise bias from differences in proteomic coverage. For the purpose of enrichment tests, we obtained the lists of differentially expressed or selected proteins as proposed by the authors of respective studies.

The synopsis graphic was created with BioRender.com

# Data availability

The computer code and results produced in this study are available in the following databases: Modelling computer scripts: GitHub (https://github.com/comp-med/protein-foundations-ubk-olink-50k). Results can be obtained from https://omicscience.org/apps/prot_foundation/.

The source data of this paper are collected in the following database record: biostudies:S-SCDT-10_1038-S44320-025-00158-6.

## Peer review information

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

## Acknowledgements

The authors acknowledge the Scientific Computing of the IT Division at the Charité—Universitätsmedizin Berlin for providing computational resources that have contributed to the research results reported in this paper (https://www.charite.de/en/research/research_support_services/research_infrastructure/science_it/#c30646061). This work was supported by the DZHK (German Centre for Cardiovascular Research) and the BMBF (German Ministry of Education and Research) to CL (Grant Number: 81X2100281), as well as by the DFG (German Research Foundation) to KD (Walter Benjamin Fellowship, Grant Number: 547107463). Co-funded by the European Union (ERC, GenDrug, 101116072) to MP. Views and opinions expressed are, however, those of the author(s) only and do not necessarily reflect those of the European Union or the European Research Council. Neither the European Union nor the granting authority can be held responsible for them. The funders had no role in the study design, data collection and analysis, the decision to publish or the preparation of the manuscript. This work was supported by the de.NBI Cloud within the German Network for Bioinformatics Infrastructure (de.NBI) funded by the German Federal Ministry of Education and Research (BMBF) (031A532B, 031A533A, 031A533B, 031A534A, 031A535A, 031A537A, 031A537B, 031A537C, 031A537D, and 031A538A).

## Author contributions

**Maik Pietzner**: Conceptualisation; Data curation; Formal analysis; Supervision; Funding acquisition; Investigation; Visualisation; Methodology; Writing—original draft; Writing—review and editing. **Carl Beuchel**: Resources; Data curation; Software; Visualisation; Methodology; Writing—review and editing. **Kamil Demircan**: Formal analysis; Funding acquisition; Visualisation; Methodology; Writing—review and editing. **Julian Hoffmann Anton**: Resources; Software; Visualisation; Writing—review and editing. **Wenhuan Zeng**: Resources; Software; Visualisation; Methodology; Writing—review and editing. **Werner Römisch-Margl**: Resources; Software; Visualisation; Writing—review and editing. **Summaira Yasmeen**: Data curation; Formal analysis; Writing—review and editing. **Burulça Uluvar**: Resources; Data curation; Software; Writing—review and editing. **Martijn Zoodsma**: Resources; Data curation; Writing—review and editing. **Mine Koprulu**: Resources; Data curation; Writing—review and editing. **Gabi Kastenmüller**: Resources; Software; Supervision; Funding acquisition; Project administration; Writing—review and editing. **Julia Carrasco-Zanini**: Conceptualisation; Resources; Software; Investigation; Methodology; Writing—review and editing. **Claudia Langenberg**: Conceptualisation; Funding acquisition; Visualisation; Methodology; Writing—original draft; Project administration; Writing—review and editing.

Source data underlying figure panels in this paper may have individual authorship assigned. Where available, figure panel/source data authorship is listed in the following database record: biostudies:S-SCDT-10_1038-S44320-025-00158-6.

## Funding

## Disclosure and competing interests statement

The authors declare no competing interests.

# Expanded View Figures

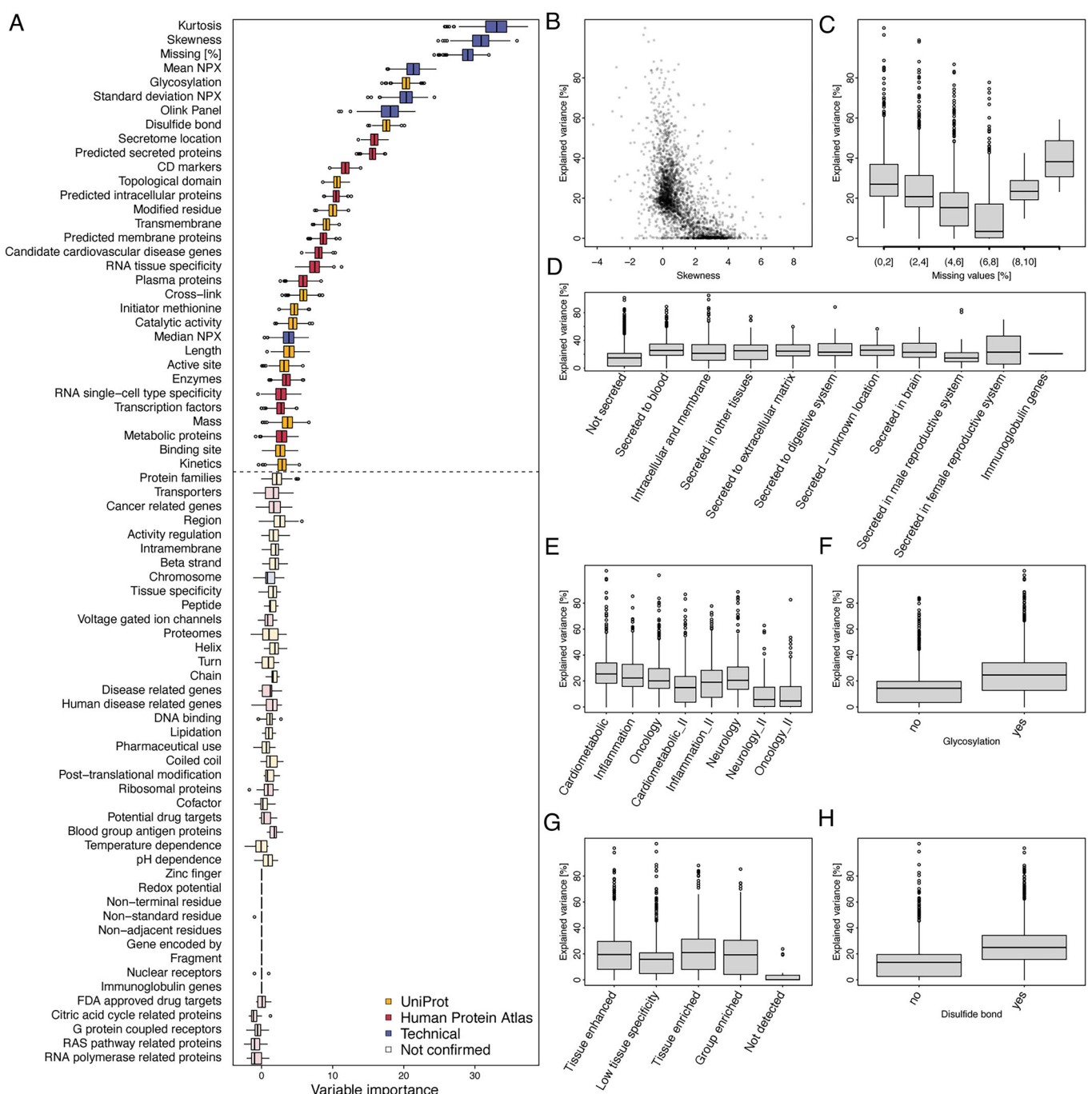

**Figure EV1. Summary of protein and assay characteristics associated with the variance explained achieved in plasma protein levels.**

(A) Variable importance of protein and assay characteristics based on Boruta feature selection, predicting the variance explained achieved for each protein target. Boxplots indicate the distribution of the variable importance across 500 iterations. Darker colours indicate features passing corrected statistical significance ($p < 0.01$). Analysis included a total of 2853 protein targets for which at least one feature that explained variation in plasma levels could be identified. Boxplots were drawn using default options: lower whiskers = 25th percentile − 1.5 x interquartile range; upper whiskers = 75th percentile + 1.5 x interquartile range; centre = 50th percentile (median); lower box bound = 25th percentile; upper box bound = 75th percentile; minima and maxima represent the most extreme values and are plotted as outliers if exceeding whiskers. (B) Scatterplot opposing the skewness of individual plasma protein distributions with the variance explained. (C–H) Variance explained according to different criteria deemed important by the Boruta feature selection. Displayed and explained variance values for a total of 2853 protein targets for which at least one feature that explained variation in plasma levels could be identified. Boxplots were drawn using default options: lower whiskers = 25th percentile - 1.5 x interquartile range; upper whiskers = 75th percentile + 1.5 x interquartile range; centre = 50th percentile (median); lower box bound = 25th percentile; upper box bound = 75th percentile; minima and maxima represent the most extreme values and are plotted as outliers if exceeding whiskers.

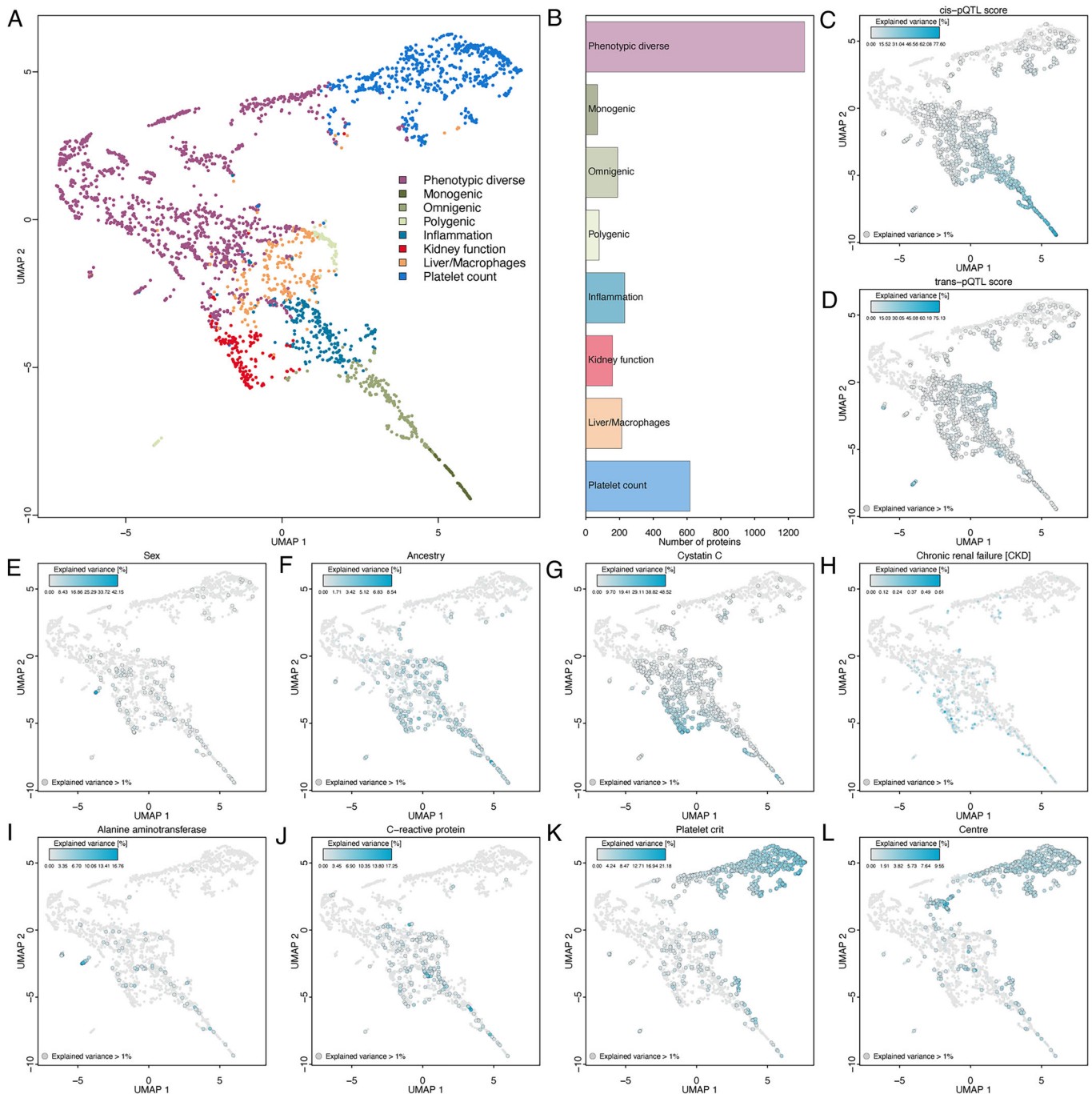

**Figure EV2. Major foundations of plasma protein variation.**

(A) Uniform manifold approximation and projection (UMAP) mapping of the variance explained matrix across 2853 protein targets for which we identified at least one feature explaining the variance in plasma levels. Each protein has been assigned a cluster based on k-means clustering and is coloured accordingly. (B) Number of protein targets included in each cluster. (C–L) Same UMAP plot but coloured according to the variance explained by the factor given on top of each plot. Proteins with strong contributions (>1%) are highlighted. pQTL protein quantitative trait loci.

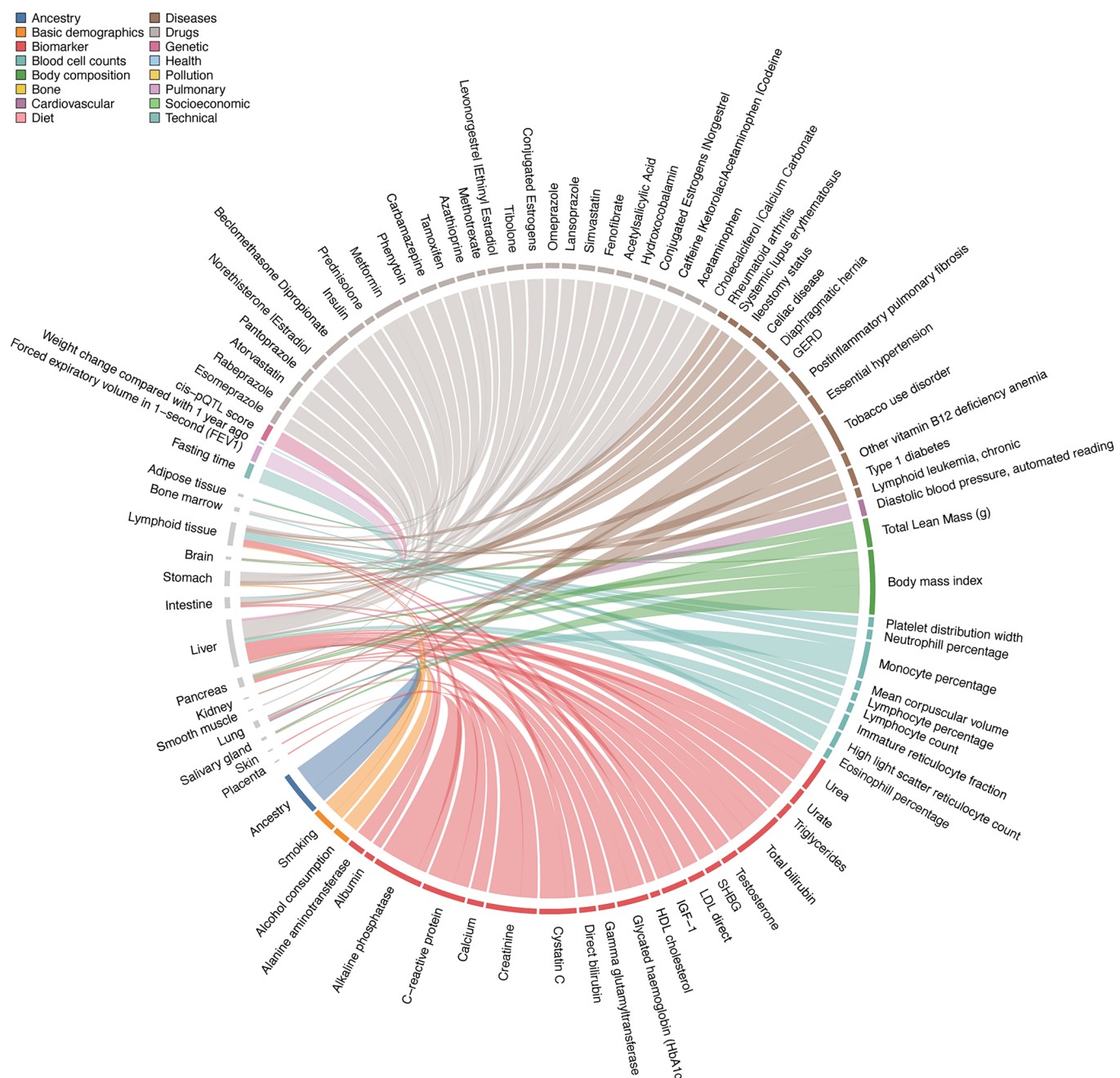

**Figure EV3. Chord diagram of phenotype-associated protein signature enrichment among protein-coding genes with enhanced tissue expression.**

Each line represents a significant enrichment ($p < 6.9 \times 10^{-6}$) of proteins associated with a given participant characteristic among protein-coding genes with enhanced expression in a given tissue. Enhanced expression estimates were derived from the Human Protein Atlas. Corresponding statistics can be found in Dataset EV7. Colouring was done according to phenotype categories as introduced in Fig. 1 in the main text.

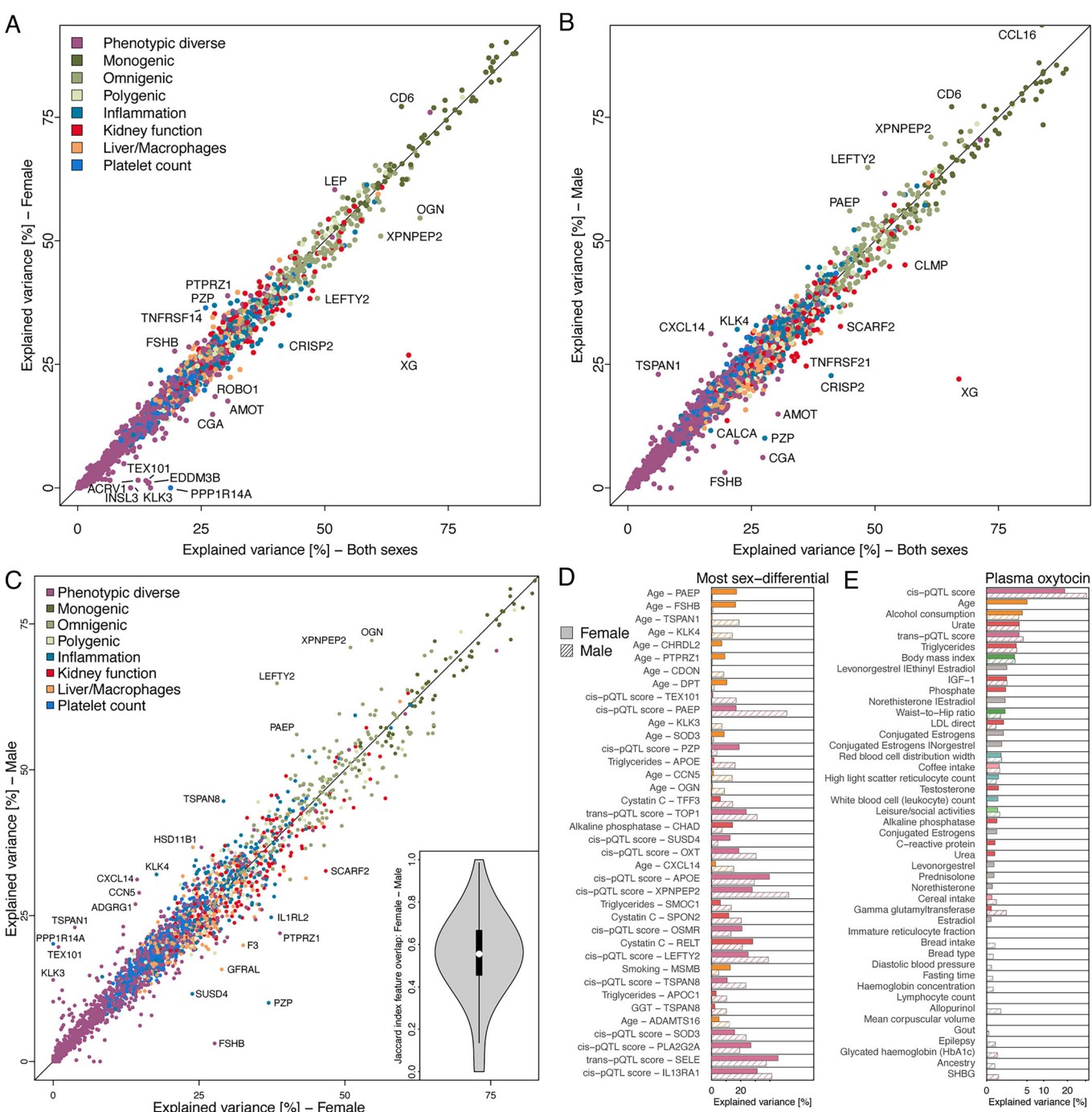

**Figure EV4.  Summary of sex-differential feature selection and contribution.**

(A, B) Scatterplots comparing the variance explained achieved for plasma protein levels among the entire UK Biobank population (x-axis) compared to what was achieved in females (left) and males (right) alone. Proteins that deviated the most (>4 s.d.) were annotated. (C) Comparison of cumulative variance explained when stratifying the UK Biobank population by sex (n = 23,601 females, n = 20,055 males). Proteins are coloured by cluster assignments as introduced in Fig. 1. Proteins with strong differences are annotated with gene names. The inlet depicts the distribution of the Jaccard index of overlapping features for the same protein across the sexes. Boxplot was drawn using default options: lower whiskers = 25th percentile − 1.5 × interquartile range; upper whiskers = 75th percentile + 1.5 × interquartile range; centre = 50th percentile (median); lower box bound = 25th percentile; upper box bound = 75th percentile; minima and maxima represent the most extreme values and are plotted as outliers if exceeding whiskers. (D) Protein—Feature combinations with significant evidence for sex-differential effects (p < 9.0 × 10$^{-7}$). Only combinations with a difference of more than 10% are shown. Sex-differential effects were assessed using an interaction term in a linear regression model. (E) Individual variance explained estimates for plasma oxytocin levels by sex, ordered by the variance explained among females. Proteins in b and c are coloured according to feature categories as introduced in Fig. 1. Variance explained was derived from a multivariable linear regression model as the partial $R^2$.

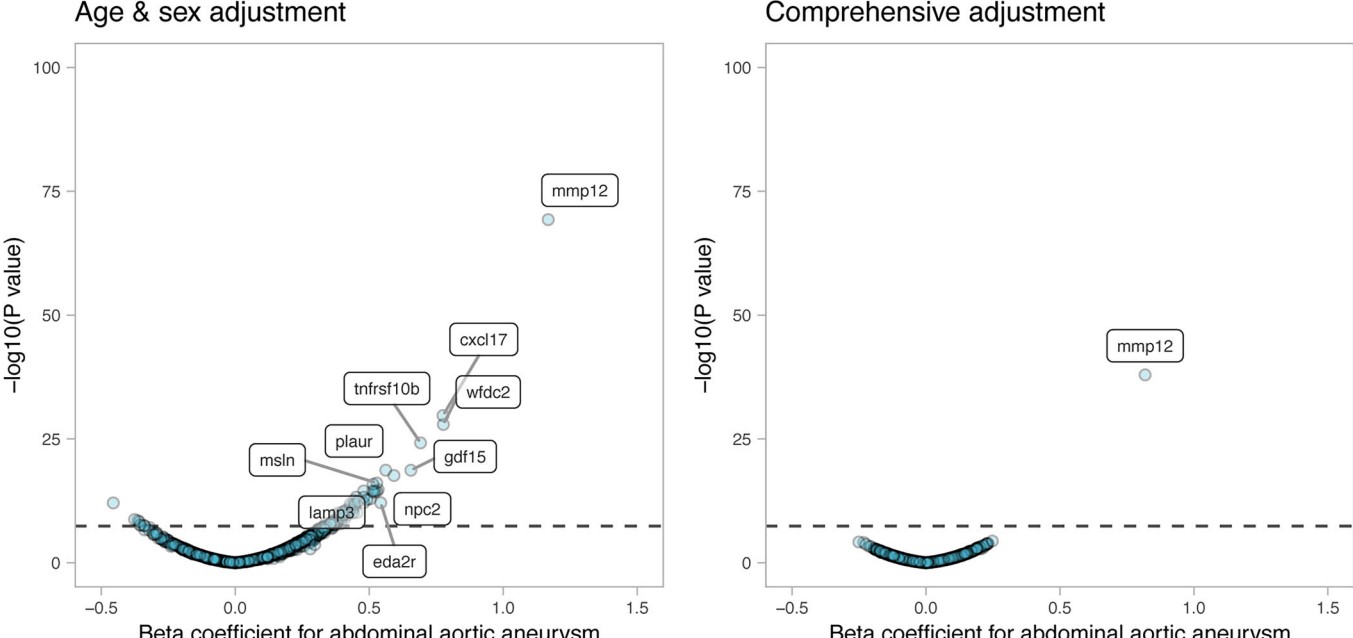

**Figure EV5.  Plasma protein biomarkers associated with the onset of abdominal aortic aneurysm.**

Volcano plots for protein–abdominal aortic aneurysm associations, with adjustment for age and sex (left panel), or comprehensive adjustment based on results from the feature selection algorithm for each protein target (right panel). Association statistics were derived from Cox-proportional hazard models.

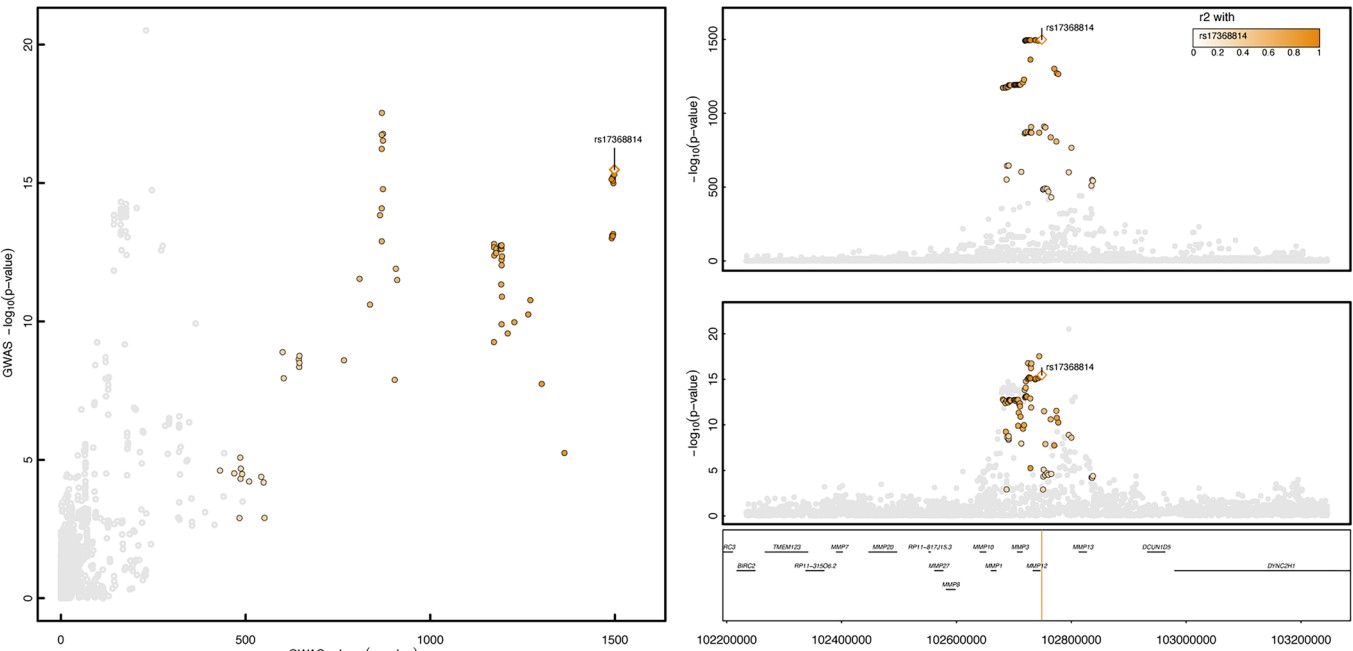

**Figure EV6.   Genetic variation at the *MMP12* locus is associated with the onset of abdominal aortic aneurysm.**

Regional association plot of the *MMP12* locus for MMP12 protein plasma levels (top) and abdominal aortic aneurysm (AAA, bottom). One single-nucleotide polymorphism (rs17368814) has been prioritised as a shared genetic signal. Summary statistics from logistic regression models for AAA are publicly available from the AAAgen consortium and summary statistics from linear regression models for MMP12 plasma levels are based on UK Biobank. Colouring indicates linkage disequilibrium to the lead variant of the same colour code. Probability for a shared genetic signal (PP-H$_4$) is given as PP$_{\text{rs17368814}}$ = 97.5%.

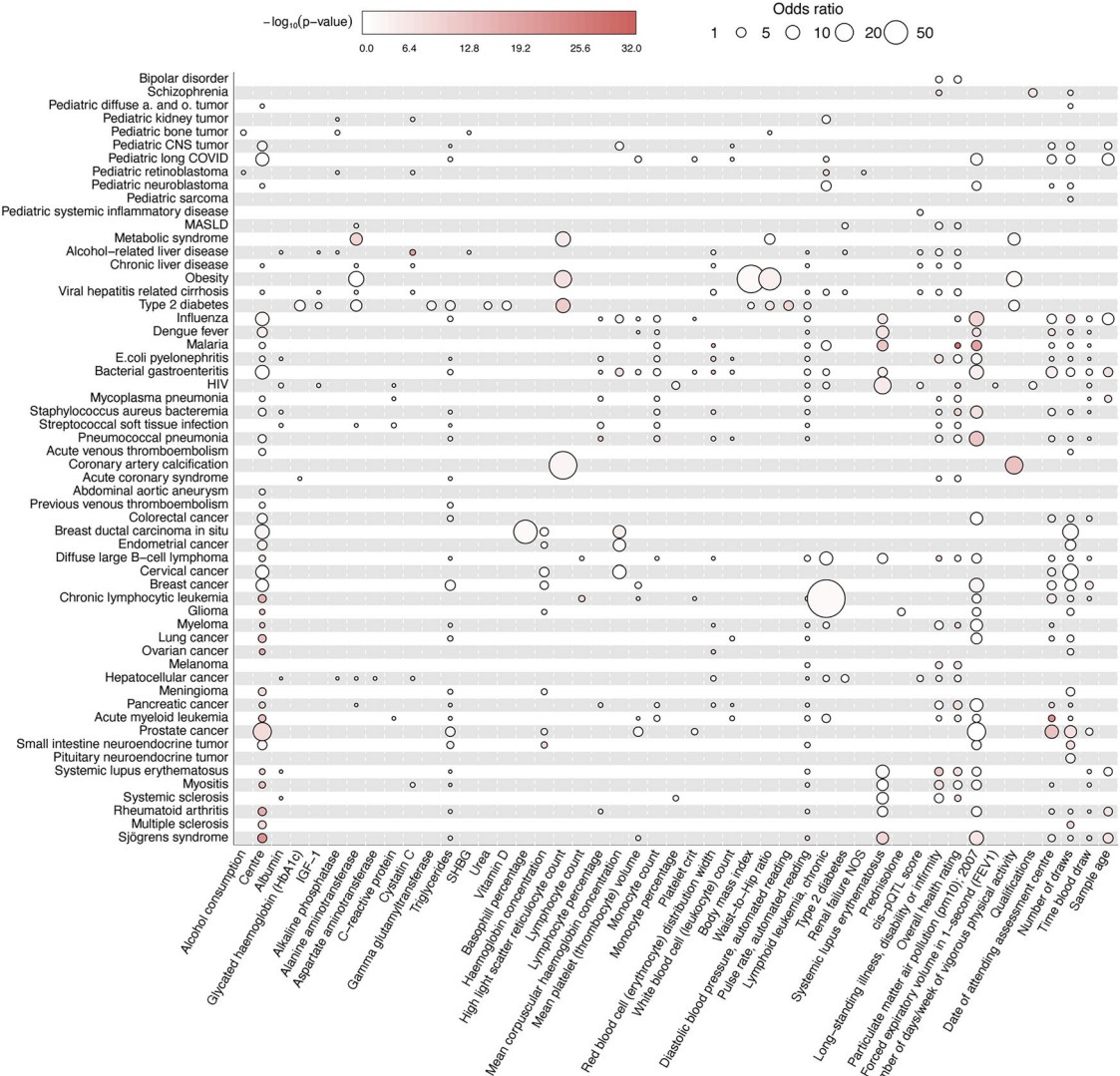

**Figure EV7. Phenotype enrichment among differentially expressed plasma proteins between a random set of controls and different patient groups from the blood protein atlas section of the Human Protein Atlas.**

Each row refers to a protein signature significantly differential (corrected $p$ value <0.05 and |log-fold change| >0.5) in the plasma of diseased patients. Each column refers to phenotype-associated protein signature from the protein atlas that was significantly ($p < 4.7 \times 10^{-6}$) enriched among the respective disease–protein signatures. Significant findings are highlighted by dots, with colour representing transformed $p$ values and size reflecting odds ratios. Enrichment was done using Fisher's exact test.

