## [Peer Review File · Molecular Systems Biology]

Machine learning-guided deconvolution of plasma protein levels

Maik Pietzner, Carl Beuchel, Kamil Demircan, Julian Hoffmann Anton, Wenhuan Zeng, Werner Römisch-Margl, Summaira Yasmeen, Burulça Uluvar, Martijn Zoodsma, Mine Koprulu, Gabi Kastenmüller, Julia Carrasco-Zanini, and Claudia Langenberg

Corresponding author(s): Maik Pietzner (maik.pietzner@bih-charite.de) , Claudia Langenberg (claudia.langenberg@qmul.ac.uk)

Review Timeline:

Submission Date:	14th Apr 25
Editorial Decision:	30th May 25
Revision Received:	15th Jul 25
Editorial Decision:	10th Sep 25
Revision Received:	11th Sep 25
Accepted:	22nd Sep 25

Editor: Poonam Bheda

Transaction Report:

30th May 2025

Manuscript Number: MSB-2025-13038-T

Title: Machine learning-guided deconvolution of plasma protein levels

Dear Dr. Pietzner,

Thank you for the submission of your manuscript to Molecular Systems Biology. We have now received feedback from the three reviewers who agreed to evaluate your manuscript. As you will see from the reports below, the referees acknowledge the interest of the study and are overall supportive of your work; however they also comment on multiple aspects of the manuscript that should be strengthened in a revision.

Without repeating all the comments listed below, some of the more fundamental issues raised are the following:

- Lack of replication in another cohort
- Imprecise language and unclear methodology
- Lack of context of the work and what is new in comparison to what you and others have previously done

All other issues raised would need to be satisfactorily addressed. Please let me know in case you would like to discuss in further detail any of the comments, I would be happy to schedule a call.

We require:

1) A .docx formatted version of the manuscript text (including legends for main figures, EV figures and tables). Please make sure that the changes are highlighted to be clearly visible. Alternatively you may choose to submit your manuscript as a LaTeX file.

4) A .docx formatted letter INCLUDING the reviewers' reports and your detailed point-by-point responses to their comments. As part of the EMBO Press transparent editorial process, the point-by-point response is part of the Peer Review File (PRF), which will be published alongside your paper.

5) A complete author checklist, which you can download from our author guidelines (<https://www.embopress.org/page/journal/17574684/authorguide#submissionofrevisions>). Please insert information in the checklist that is also reflected in the manuscript. The completed author checklist will also be part of the PRF.

6) Please note that all corresponding authors are required to supply an ORCID ID for their name upon submission of a revised manuscript.

7) It is mandatory to include a 'Data Availability' section after the Materials and Methods. Before submitting your revision, primary datasets produced in this study need to be deposited in an appropriate public database, and the accession numbers and database listed under 'Data Availability'. Please remember to provide a reviewer password if the datasets are not yet public (see <https://www.embopress.org/page/journal/17574684/authorguide#dataavailability>).

In case you have no data that requires deposition in a public database, please state so in this section as follows: "This study includes no data deposited in external repositories". Note that the Data Availability Section is restricted to new primary data that are part of this study.

8) All Materials and Methods need to be described in the main text using our 'Structured Methods' format, which is required for all research articles. According to this format, the Methods section includes a Reagents and Tools Table (listing key reagents, experimental models, software and relevant equipment and including their sources and relevant identifiers) followed by a Methods and Protocols section describing the methods using a step-by-step protocol format. The aim is to facilitate adoption of the methodologies across labs. Please upload the Reagents and Tools table as a separate document when submitting your revised manuscript. More information on how to adhere to this format as well as a downloadable template (.docx) for the Reagents and Tools Table can be found in our author guidelines:

<https://www.embopress.org/page/journal/17444292/authorguide#structuredmethods>

9) For data quantification: please specify the name of the statistical test used to generate error bars and p-values, the number (n) of independent experiments (specify technical or biological replicates) underlying each data point and the test used to calculate p-values in each figure legend. The figure legends should contain a basic description of n, p-values and the test applied. Graphs must include a description of the bars and the error bars (s.d., s.e.m.). Please provide exact p-values (in either the figure or figure legend).

10) Our journal encourages inclusion of *data citations in the reference list* to directly cite datasets that were re-used and obtained from public databases. Data citations in the article text are distinct from normal bibliographical citations and should directly link to the database records from which the data can be accessed. In the main text, data citations are formatted as follows: "Data ref: Smith et al, 2001" or "Data ref: NCBI Sequence Read Archive PRJNA342805, 2017". In the Reference list, data citations must be labeled with "[DATASET]". A data reference must provide the database name, accession number/identifiers and a resolvable link to the landing page from which the data can be accessed at the end of the reference. Further instructions are available at .

11) We replaced Supplementary Information with Expanded View (EV) Figures and Tables that are collapsible/expandable online. EV Figures should be cited as 'Figure EV1, Figure EV2' etc... in the text and their respective legends should be included in the main text after the legends of regular figures.

- Additional Tables/Datasets should be labeled and referred to as Table EV1, Dataset EV1, etc. Legends should be provided in a separate tab in case of .xls files. Alternatively, the legend can be supplied as a separate text file (README) and zipped together with the Table/Dataset file.

<https://www.embopress.org/page/journal/17574684/authorguide#expandedview>

12) Author contributions: CRediT has replaced the traditional author contributions section because it offers a systematic machine-readable author contributions format that allows for more effective research assessment. Please remove the Authors Contributions from the manuscript and use the free text boxes beneath each contributing author's name in our system to add specific details on the author's contribution. More information is available in our guide to authors.

13) Disclosure statement and competing interests: We updated our journal's competing interests policy in January 2022 and request authors to consider both actual and perceived competing interests. Please review the policy

<https://www.embopress.org/competing-interests> and update your competing interests if necessary.

14) Every published paper now includes a 'Synopsis' to further enhance discoverability. Synopses are displayed on the journal webpage and are freely accessible to all readers. They include a short stand first (maximum of 300 characters, including space) as well as 2-5 one-sentences bullet points that summarizes the paper. Please write the bullet points to summarize the key NEW findings. They should be designed to be complementary to the abstract - i.e. not repeat the same text. We encourage inclusion of key acronyms and quantitative information (maximum of 30 words / bullet point). Please use the passive voice. Please attach these in a separate file or send them by email, we will incorporate them accordingly.

Please note that these would be the final versions and changes during proofing are usually not allowed.

15) As part of the EMBO Publications transparent editorial process initiative (see our policy here:

https://www.embopress.org/transparent-process#Review_Process), Molecular Systems Biology will publish online a Peer Review File (PRF) to accompany accepted manuscripts.

In the event of acceptance, this file will be published in conjunction with your paper and will include the anonymous referee reports, your point-by-point response and all pertinent correspondence relating to the manuscript. Let us know whether you agree with the publication of the PRF and as here, if you want to remove or not any figures from it prior to publication.

Please note that the Author checklist will be published at the end of the PRF.

Molecular Systems Biology has a "scooping protection" policy, whereby similar findings that are published by others during review or revision are not a criterion for rejection. Should you decide to submit a revised version, I do ask that you get in touch after three months if you have not completed it, to update us on the status.

Yours sincerely,

Poonam Bheda, PhD
Scientific Editor
Molecular Systems Biology

Reviewer #1:

In this manuscript, Pietzner et al present a large-scale analysis of the determinants of plasma protein variation using ~3000 proteins measured in ~43,000 UK Biobank participants. By integrating over 1,700 features that describe patient and technical characteristics using machine learning based feature selection and variance decomposition, the authors identify the major modifiable and non-modifiable factors influencing protein levels. The study introduces a web portal and the r-prodente R package, which allows researchers to test protein lists for enrichment of underlying phenotypic drivers. The authors demonstrate the utility of this framework through examples including disease biomarker identification, phenotype enrichment analyses, and triangulation with genetic and drug-related data.

The work presents very valuable insights into the confounding factors of plasma proteome variation. With this data-driven approach Pietzner et al revealed that many (if not most) of the associations identified between plasma proteome levels and diseases are likely confounded by (non-disease related) patient characteristics. To us these are the most important take-aways from the manuscript and we hope that these findings will encourage the field to apply greater caution when reporting novel protein-based biomarkers for diseases.

Major comments

Replication

While the authors acknowledge the limited replicability due to the unique scale of the UK Biobank, attempting replication in another large-scale cohort could strengthen the manuscript.

It would be interesting to assess whether the variance in protein levels explained by patient characteristics correlates between UKB and another cohort. For example, the authors have previously used data from the Fenland study to identify patient characteristics influencing protein-level variance. A subset of that study includes Olink assay measurements for a limited number of proteins. The availability of patient characteristics differs between cohorts, however, even if not all features from the UKB are present in Fenland, the model could be retrained on a reduced set of shared features to enable partial replication. Furthermore, replication in another population such as FinnGen could be interesting as well, this could lead to proteins where the explained variance from a certain feature would be different from UKB.

Discussion of results in relation to previous work

The manuscript is conceptually similar to work previously published by the authors (e.g. Carrasco-Zanini et al., 2024, Nat. Metab.). While it is briefly mentioned that this is a follow-up to their earlier work, it would be important to thoroughly discuss the new methodology and findings in context of the earlier study. Additionally, it would be helpful to discuss how the presented findings impact the authors' earlier work on risk prediction models (Carrasco-Zanini et al., 2024, Nat. Med.).

The authors show that many of the previously described >500,000 protein-phenotype combinations are likely driven by confounding factors. While these findings are of very high importance to the field, it would be great if the the authors discuss their own published work on risk predictions from plasma proteins in this context. Specifically, as the authors have first-hand access to the data and analysis presented in the Nat. Med. study, it might be relatively low effort to include analysis or at least discuss how the new findings affect the validity of the identified associations between proteins and rare diseases.

Minor comments

The current documentation of the r-prodente package is rather sparse. Please expand the GitHub documentation to briefly explain the method (without only referring to the paper), specify input formats, describe expected output (column names and meanings), and clearly state the statistical test being used (i.e., Fishers exact test).

Currently, the "enrichment_protein_characteristics" function returns unadjusted p-values from Fishers exact test. Since many tests are run in typical use cases, it would be helpful to include FDR- or bonferroni-adjusted pvalues directly in the output. For example, the function used for plotting the enrichment results includes bonferroni adjustment.

Please also check that the enrichment results reported in "A protein foundation community resource" pass FDR/bonferroni adjustment and report if p-values are adjusted for multiple testing or not.

Consider renaming the "Protein Atlas" feature as it may be confused with the Proteome-Phenome Atlas or the Human Protein Atlas (which is also referenced).

In the protein atlas web app, allow users to download .csv files for each selected protein. This would make it easier to access relevant data without navigating supplementary files.

Reviewer #2:

Pietzner and his or her colleagues from Universitätsmedizin Berlin leveraged 40K+ individuals from UK Biobank to investigate the variation of plasma proteins through a comprehensive range of modifiable and non-modifiable phenotypes. The study provide a guided reference for the community to explore interested proteins for actionable factors. The overall study is interest and the findings are important; however, revisions including further analysis are need to improve the study. I hope my comments would facilitate the improvement.

Major comments

1. The manuscript distinguishes between "modifiable" and "non-modifiable" participant characteristics but does not explicitly define or categorize which phenotypes fall into these groups. This indication should be added.
2. I strongly suggest the authors perform a pre-processing by adjusting age, sex, genetic factors for each protein. Then, use the residuals to re-perform the variance analysis. As I observe many proteins were largely explained by these factors, and they might be highly correlated to other phenotypes. These results are encouraged to be included as sensitivity analysis and updated in your website.
3. The manuscript mentioned handling sex-specific diseases in Cox models but does not elaborate. How were sex-specific conditions (e.g., prostate cancer, ovarian cancer) addressed in the analysis? Were models stratified by sex, or were sex-specific diseases excluded from certain analyses?
4. Missing imputation for phenotype data using random forest should be more clearly elaborate, e.g., did you only perform imputation to 70% data and subsequently perform feature selection? Did you impute all features together? Or impute one by one?
5. With >1700 features, how was multicollinearity addressed during LASSO-based feature selection? Did the authors consider the issue of correlated predictors (e.g., BMI and waist-to-hip ratio)?
6. The average variance explained by identified factors is 23.7%. However, a substantial proportion of variance remains unexplained. Authors should add discussion on potential sources of unaccounted variation (e.g., unmeasured environmental factors, protein turnover rates, or assay limitations)?
7. The lower explained variance in British African and South Asian groups is attributed partly to genetic differences. However, the sample sizes for these groups are small (n=849 and 823). Were sensitivity analyses (e.g., subsampling Europeans to match sample sizes) performed to confirm that the differences are not primarily due to reduced statistical power?
8. The knowledge graph integrates self-reported medication data, which may be confounded by indication. Were comorbidities or concurrent treatments adjusted for in drug-protein associations? How robust are the inferred drug-target links?

Reviewer #3:

Summary:

The authors analyzed plasma proteins in UKBB individuals to identify factors responsible for variations in their plasmatic concentrations. They assessed numerous non-genetic features, patient's characteristics, protein-specific characteristics and previously identified genetic variants (cis- and trans-pQTLs), and tried to quantify the effect of all these variables on plasma protein levels, assessing such effects for each sex and across three different population of different ancestry. Next, the authors try to identify plasma proteins that could be used as disease biomarkers. The knowledge-graph provided by the authors is a great way to gather their results into an easy-to-use summary for large-scale studies.

Considerations:

The work seems interesting, many results are in line with previous findings, but the overall writing is unprecise and make the whole work and the underlying methodology very opaque.

As of now, it is really hard for me to produce an clear and informed feedback about the scientific content of the manuscript, as the methods used are unclear, unprecise or not detailed in the manuscript. An additional writing effort from the authors is needed so that the quality of the scientific content can be vouched for.

Nevertheless, while I think this manuscript requires some changes to make it clear, the underlying hypotheses and questions appear sound and interesting. I'd like to be involved in a second round of review, once my following points will have be included.

Please note that readers will probably approach this work by reading the abstract & the results, and that i) a clearer description

of what was previously done, including summary of relevant previous/referenced works when pointing toward a specific methodology/dataset, ii) explained, limited and consistent jargon usage, and iii) shorter sentences, would greatly improve the readability of this article.

For examples:

- This work is based on the field of population genetics, which uses the word "variation" in a very specific way, i.e. to design genetic variants.

While the authors used this word appropriately multiple times, it sometimes refer to plasma protein level variations: e.g. line 164: "The single strongest contributor to ancestral variation was the contribution of cis- and trans-pQTLs (Supplementary Tab. 7), likely explained by a combination of different reasons.", a sentence which also integrates references to genetic variations (cis- and trans- pQTLs) as contributors of the mentioned "ancestral variations", which can refer to both ancestral genetic variants, or to (what I believe the authors want to mention) ancestry-specific plasma protein explained variance. If so, please refer to these different concepts precisely, consistently and appropriately along the manuscript.

- Descriptors such as "ancestral regional lead variant" only make the reading more complex. Might be replaced with "lead genetic variant with ancestry-specific effect", which is indeed longer, but clearer and more precise. The "regional" descriptor is redundant with the "lead", as a lead variant is representative of several variants in a given region.

- line 203 "To transcend statistical mapping of protein - disease associations that are unlikely to inform prioritisation of biologically or clinically relevant insights, we next built a framework designed to provide the foundation for robust biomarker identification in biobank-scale studies" -> sentence is too long, "prioritising insights" is very vague, "statistical mapping of [...] associations" doesn't make sense...

Methods are incomplete, no mention of how QTLs score were obtained, while they are used throughout the manuscript.

** (non-exhaustive) ** corrections:

Results:

67-69 : "Protein targets thereby segregated into distinct clusters according to major influences on human health and but also pre-analytical variation activating platelets"?

140: extend -> extent

151, 190,

154: "differential or -specific" -> "differential or specific"

155: "and the sexes"- > "and sexes"

167-170: "We obtained evidence that differential effects of the same variant across ancestral groups most commonly accounted for differences and included 134 cis-pQTLs, almost all (n=130) of which could not be explained by different causal variants across ancestries." -> most commonly accounted for what differences? Sentence should be cut in half to improve readability, and the second part of the sentence reshaped to improve its clarity.

184: "pairs" -> "pairs"

190: "ancestral-lead signals" -> ancestry-specific lead signal (ancestry-specific association signal between cis-pQTL and plasma protein levels)?

197: "trans-ancestral" -> "cross-ancestral", to avoid using confusion with trans-pQTLs which are also mentioned in the paper.

198, 199: "ancestral-specific lead signal" -> ancestry-specific lead signal

205: "Among the 60,334 significant disease associations ($p < 4.4 \times 10^{-8}$) observed here and reported in other studies, we observed a more than 39-fold drop after regressing out foundations of plasma protein levels for 390 incident diseases)" -> "a 39-fold drop", and what are the foundations that are regressed out? No mention whatsoever of such "foundations" are made in the manuscript. A dedicated section in the Methods part of the manuscript is needed, detailing model used, included covariates... And why would you regress out all explaining covariates to identify biomarkers?

211: ">80% (n=1,300) showed considerable effect attenuation of {greater than or equal to}20%, suggesting residual confounding" -> 20% of what? residual confounding of what by what?

222: "Less-established links with persistent considerable effect size included a 2.5-fold increased risk" -> persistent what?

228: "We further obtained evidence for a potentially causal effect of high plasma levels of MMP12 on the risk of abdominal aortic aneurysm (hazard ratio=1.15; p-value=3.3x10-16 230) by establishing a shared genetic signal (Posterior probability: rs17368814=97.5%, Supplementary Fig. 6)." -> Is it just not co-localization of signal? Claims of potential causality require more evidence, or at least a potential mechanistic explanation of causality.

232: "Prioritisation of biomarker associations has previously been proposed using genetic imputation 14" -> what prioritisation of biomarker associations? The article referenced herein does not make any mention of prioritisation. Is it not only about identifying potential disease biomarkers among the assessed proteins?

237: "This included not only strongly differing effect estimates, and hence power to stratify people at risk, but also discordant results whether considering cis- or trans pQTLs for imputing plasma protein levels " -> size effect estimates? What does "hence power to stratify people at risk" mean?

Methods:

463-464: please briefly summarize how the pQTLs were mapped.

479: please briefly report the filters used.

494: "We modelled categorical variables as random effects" -> why?

505-506: "This test was only done for variables not specific to only of the strata" -> a word is probably missing somewhere.
528: "silhouette coefficient" -> please briefly describe what it is.
535: "We declared ancestral-specific effects, if..." -> "We declared an effect as ancestry-specific if..."
539-540 "ancestries and computed the explained variance as well as tested for a significant interaction effects." -> interactions with what?
541 : "ancestral-differential explained variance" ->
557: "maximise the advantages of the knowledge graph" -> what advantages?
568: "Cox proportional hazards models" -> please briefly summarize what results such models produce
572: "cis-pQTL scores, and trans-pQTL scores" -> how were they computed?
580: "Schoenfeld residuals" -> please briefly summarize what they are
586: "tissue-specifically expressed proteins" -> proteins with tissue-specific expression patterns
587: "were enriched among those best explained by the same major biological factors." "were enriched among the proteins best explained by the same major biological factors", and what do you mean by "the same major biological factors"? Where is this similarity coming from?
591 -> same question as for line 587

Figures:

F1b: not referenced in the text.

F1c: very hard to read, categories are not distinguishable for each protein, not sure of the interest of this figure in the main body of this manuscript.

F1d : omingenic -> omnigenic

SF1a: "darker colours indicate features passing statistical significance" -> cant see any darker colour?

SF1a: the authors assess the explained variance of plasma proteins using protein characteristics extracted and/or computed from UniProt & the Human Protein Atlas. I'm not sure that

SF2c-i : please homogeneize the colorbar used for the explained variance, as it changes with each plot, it would improve the readability of the figure.

SF3e -> cant see any darker colour?

SF4e: not referenced in text

Dear Poonam,

We are grateful for the opportunity to provide a revised version of our work, which we think has clearly benefited from the valuable input from the two reviewers.

Please find our point-by-point response to reviewers below, in which we outline our responses and changes made to address all questions, suggestions and helpful comments that were raised.

In detail, we now include a rigorous replication of our results in another cohort and provide evidence of good coherence even if different platforms and participant characteristics were used to explain plasma protein levels. The comparison clearly demonstrated the substantial added value of the present study in comparison to what we and others have done previously. We further improved the writing to ease accessibility and clarified methods according to the points raised by the reviewers.

We note that implementing suggested updates to the analytical approach, i.e., variance estimation, meant that we needed to rerun all analyses. We took this opportunity to update three critical data resources, namely the assessment of drug intake and disease onset (e.g., >100 novel drugs not considered in the previous version) as well as integration of findings reported in the GWAS Catalog. These updates clearly strengthened the manuscript (e.g., we now identify DDX58 as an additional potential marker of target engagement for Interferon beta-1a), but also meant that point estimates for the explained variance had to be updated throughout the manuscript.

We hope you agree that these changes and the new results strengthen our work substantially and hope that this is now acceptable for publication in *Molecular Systems Biology*.

Yours sincerely,

Maik Pietzner and Claudia Langenberg, on behalf of the team

Reviewer #1:

In this manuscript, Pietzner et al present a large-scale analysis of the determinants of plasma protein variation using ~3000 proteins measured in ~43,000 UK Biobank participants. By integrating over 1,700 features that describe patient and technical characteristics using machine learning based feature selection and variance decomposition, the authors identify the major modifiable and non-modifiable factors influencing protein levels. The study introduces a web portal and the *r-prodente* R package, which allows researchers to test protein lists for enrichment of underlying phenotypic drivers. The authors demonstrate the utility of this framework through examples including disease biomarker identification, phenotype enrichment analyses, and triangulation with genetic and drug-related data.

The work presents very valuable insights into the confounding factors of plasma proteome variation. With this data-driven approach Pietzner et al revealed that many (if not most) of the associations identified between plasma proteome levels and diseases are likely confounded by (non-disease related) patient characteristics. To us these are the most important take-aways from the manuscript and we hope that these findings will encourage the field to apply greater caution when reporting novel protein-based biomarkers for diseases.

We thank the reviewer for the time taken to carefully review our work. We are delighted to hear that our work provides a valuable resource for the field.

Major comments

Replication

While the authors acknowledge the limited replicability due to the unique scale of the UK Biobank, attempting replication in another large-scale cohort could strengthen the manuscript.

It would be interesting to assess whether the variance in protein levels explained by patient characteristics correlates between UKB and another cohort. For example, the authors have previously used data from the Fenland study to identify patient characteristics influencing protein-level variance. A subset of that study includes Olink assay measurements for a limited number of proteins. The availability of patient characteristics differs between cohorts, however, even if not all features from the UKB are present in Fenland, the model could be retrained on a reduced set of shared features to enable partial replication. Furthermore, replication in another population such as FinnGen could be interesting as well, this could lead to proteins where the explained variance from a certain feature would be different from UKB.

***R1 response 1** We followed the recommendation of the reviewer and now provide a detailed replication study using our previously published work (page 4, line 96-106), on top of the already implemented testing and validation step within UK Biobank (which we now clarify in the updated manuscript). Despite differences in proteomic technology and cohort characteristics, we observed significant correlation in the overall amount of variance explained for 1968 overlapping targets ($r=0.28$; $p\text{-value}\leq 6.7\times 10^{-37}$) that improved to 0.46*

once restricting to 699 protein targets for which measurements agree well across platforms. We further provide evidence that the expanded phenotypic spectrum in our present study significantly contributed to the higher explained variance.

We further appreciate that exploration in another cohort would be interesting to, for example, highlight novel determinants of plasma protein levels. However, considerable differences in available phenotypic data, for example FinnGen has little information on comprehensive clinical chemistry assessment that considerably explained plasma protein variation, would not allow distinguishing between a truly novel effect or missing adjustment for important confounders.

We now highlight the need to expand efforts such as ours to other populations to enhance our understanding of the plasma proteome (p13, lines 367-377). Notably, our comprehensive analysis of important strata within UK Biobank, i.e., the sexes and ancestral groups, did only provide limited evidence for strong differences in the explained variance by participant characteristics.

Discussion of results in relation to previous work

The manuscript is conceptually similar to work previously published by the authors (e.g. Carrasco-Zanini et al., 2024, Nat. Metab.). While it is briefly mentioned that this is a follow-up to their earlier work, it would be important to thoroughly discuss the new methodology and findings in context of the earlier study. Additionally, it would be helpful to discuss how the presented findings impact the authors' earlier work on risk prediction models (Carrasco-Zanini et al., 2024, Nat. Med.).

R1 response 2 *We appreciate the similarity to our previous work. While the methodology is comparable, the major distinction is the scale and breadth of participant information available in UK Biobank compared to our previous work. This is most relevant for the inclusion of drug- and disease-related information ($n > 1,764$ feature) that we now demonstrate to account for most of the additionally explained variance (p4, line 103-106). This further included putative drug target engagement markers or actionable biomarkers such as MMP12 for abdominal aortic aneurysm. We now provide a detailed benchmarking with our previous work (see previous comment) and discuss implications of the present work on protein biomarker models derived in Carrasco-Zanini et al., 2024, Nat. Med. (see below).*

The authors show that many of the previously described >500,000 protein-phenotype combinations are likely driven by confounding factors. While these findings are of very high importance to the field, it would be great if the the authors discuss their own published work on risk predictions from plasma proteins in this context. Specifically, as the authors have first-hand access to the data and analysis presented in the Nat. Med. study, it might be relatively low effort to include analysis or at least discuss how the new findings affect the validity of the identified associations between proteins and rare diseases.

R1 response 3 *We followed the reviewer's advice and have now systematically tested to what extent our prior work identifying protein predictors for disease onset might be impacted by performing a series of additional analysis to identify possible residual risk factor*

associations or obvious participant characteristics that would explain predictive performance of protein signatures.

Among 67 protein models that outperformed standard clinical models, predictive protein signatures for a third showed evidence for enrichment of phenotype-associated signatures detected in the present work ($p < 3.3 \times 10^{-6}$; new Tab. EV15). Those results did not reveal evidence for missing inclusion of informative features but rather segregated into two categories demonstrating the power of sparse feature selection. Firstly, protein signatures were enriched for characteristics that likely shared tissue damage as an underlying reason (e.g., intake of proton pump inhibitors being enriched among proteins selected to predict the onset of benign neoplasm of the stomach). Secondly, protein signatures likely represented refinements of established risk factors that had already been considered during model development (e.g., smoking being enriched among proteins selected for COPD).

The latter was even more apparent when we performed enrichment analysis among 66 proteins selected ≥ 5 across all outcomes. The 21 phenotypic signatures included subjective proxies of frailty and socioeconomic status (e.g., overall health rating: odds ratio: 5.38; p -value $< 1.9 \times 10^{-6}$) but also the presence of specific diseases such as myocardial infarction (odds ratio: 71.2; p -value $< 1.9 \times 10^{-8}$) for which associated proteins, such as NT-proBNP, may indicate disease severeness or poor recovery.

We note that those results also provide evidence for more objective assessment of otherwise hard to define personal circumstances, such as socioeconomic status, by measuring plasma protein levels. We incorporated this assessment in the revised version of the manuscript (p12, lines 332-336).

Minor comments

The current documentation of the r-prodente package is rather sparse. Please expand the GitHub documentation to briefly explain the method (without only referring to the paper), specify input formats, describe expected output (column names and meanings), and clearly state the statistical test being used (i.e., Fishers exact test).

R1 response 4 *We implemented the helpful suggestions by the reviewer and updated the documentation of the R package accordingly (see <https://github.com/comp-med/r-prodente>).*

Currently, the "enrichment_protein_characteristics" function returns unadjusted p-values from Fishers exact test. Since many tests are run in typical use cases, it would be helpful to include FDR- or bonferroni-adjusted p-values directly in the output. For example, the function used for plotting the enrichment results includes bonferroni adjustment.

R1 response 5 *We implemented correction for multiple testing in the enrichment_protein_characteristics() function as suggested.*

Please also check that the enrichment results reported in "A protein foundation community resource" pass FDR/bonferroni adjustment and report if p-values are adjusted for multiple testing or not.

R1 response 6 We apologize for the missing information and added a reference that all reported associations passed Bonferroni adjustment for multiple testing (page 9, lines 247).

Consider renaming the "Protein Atlas" feature as it may be confused with the Proteome-Phenome Atlas or the Human Protein Atlas (which is also referenced).

R1 response 7 We followed the helpful recommendation of the reviewer and now refer to the paper rather than a protein atlas.

In the protein atlas web app, allow users to download .csv files for each selected protein. This would make it easier to access relevant data without navigating supplementary files.

R1 response 8 We thank the reviewer for the advice and implemented the suggested functionality along with other updates.

Reviewer #2:

Pietzner and his or her colleagues from Universitätsmedizin Berlin leveraged 40K+ individuals from UK Biobank to investigate the variation of plasma proteins through a comprehensive range of modifiable and non-modifiable phenotypes. The study provide a guided reference for the community to explore interested proteins for actionable factors. The overall study is interest and the findings are important; however, revisions including further analysis are need to improve the study. I hope my comments would facilitate the improvement.

We thank the reviewer for their valuable time and comments on our manuscript.

Major comments

1. The manuscript distinguishes between "modifiable" and "non-modifiable" participant characteristics but does not explicitly define or categorize which phenotypes fall into these groups. This indication should be added.

R2 response 1 We apologize for the missing definition and now clearly define which features we refer to as non-modifiable (i.e., age, sex, ancestry, and common genetic variants) and modifiable (the remaining apart from technical characteristics) (page 5, lines 122-123). We acknowledge that this distinction should be considered pragmatic, given that, for example, many chronic diseases are not yet modifiable.

2. I strongly suggest the authors perform a pre-processing by adjusting age, sex, genetic factors for each protein. Then, use the residuals to re-perform the variance analysis. As I observe many proteins were largely explained by these factors, and they might be highly correlated to other phenotypes. These results are encouraged to be included as sensitivity analysis and updated in your website.

R2 response 2 We appreciate that age and sex, and possibly even genetic factors, are frequently used as standard covariates in statistical analysis due to correlation with outcomes of interest. Here, we accounted for such effects in multiple ways.

Firstly, the final variance estimation is based on a full linear model that considers all selected variables at the same time, including age, sex, and genetic factors (if selected), which is statistically equivalent to regressing out effects first. Most importantly, our approach retains interpretability by providing variance estimates most closely linked to the actual blood measurements rather than residuals.

Secondly, participants age (or sex) is highly correlated with the onset of many diseases and drug intake, and hence correcting for age before considering such factors might underestimate the variance explained by these important clinical variables. Instead, our feature selection approach is free to choose on whether participants age is more informative compared to other characteristics. Further, considering age and sex (if selected as relevant) along with other selected features in a multivariable linear model ensures proper modelling of the covariance structure among features, including the correlation between age/sex and other selected features.

Thirdly, the contribution of genetic factors is almost exclusively independent from all the >1,700 participant characteristics, as we also demonstrate when associating those with disease risk.

To further address the concern raised by the reviewer about the correlation structure among participant characteristics, we have now adopted a different statistical approach to estimate the partial explained variance specifically modelling the correlation structure among selected features as implemented in the R package *heplots* (p18, lines 519-525). We note that this led to slightly varying variance estimates as compared to the previous iteration of the results.

3. The manuscript mentioned handling sex-specific diseases in Cox models but does not elaborate. How were sex-specific conditions (e.g., prostate cancer, ovarian cancer) addressed in the analysis? Were models stratified by sex, or were sex-specific diseases excluded from certain analyses?

R2 response 3 We apologize for the missing information, and now report that for sex-specific diseases, Cox-models were only performed in the relevant sex (p19, lines 564-565).

4. Missing imputation for phenotype data using random forest should be more clearly elaborate, e.g., did you only perform imputation to 70% data and subsequently perform feature selection? Did you impute all features together? Or impute one by one?

R2 response 4 We appreciate the importance of this section and now elaborate in more detail, on how imputation was performed. Briefly, we performed imputation only once on the entire data set before splitting in feature selection and variance estimation data set. To derive maximally informative models, all participant characteristics but no protein values were considered for imputation (page 16, lines 455-458).

5. With >1700 features, how was multicollinearity addressed during LASSO-based feature selection? Did the authors consider the issue of correlated predictors (e.g., BMI and waist-to-hip ratio)?

R2 response 5 The LASSO framework has been specifically developed to address multicollinearity during feature selection, but we and others observed instabilities in feature selection at two levels: 1) the type of features selected for a given protein, and 2) instability in the selection of features likely representing similar participant across proteins. We implemented bootstrapping for 1) to achieve a list of most robustly selected features for a given protein, and 2) pruned participant characteristics before feature selection using a correlation threshold of 0.85 to identify cluster of highly correlated features. We chose a threshold of 0.85 based on interrogating a correlation network of participant characteristics that grouped into biological plausible subnetworks. For example, BMI and waist-to-hip ratio were only moderately correlated in our study ($r=0.42$) and hence both were taken forward for feature selection. We note that BMI was significantly more often selected ($n=839$ vs 706) and explained on average more variance (6.94% vs 0.14%) compared to waist-to-hip ratio in line with the latter being a mere refinement of overall fat distribution.

6. The average variance explained by identified factors is 23.7%. However, a substantial proportion of variance remains unexplained. Authors should add discussion on potential sources of unaccounted variation (e.g., unmeasured environmental factors, protein turnover rates, or assay limitations)?

R2 response 6 The reviewer refers to an important finding of our study that we now discuss in more detail (p12, lines 338-346). Briefly, based on our findings, we hypothesize that unmeasured factors may explain a high amount of variance only for selected protein targets and that technical factors likely account for the missing explained variance for most. For example, we obtained conclusive evidence that proteins with better QC measures had higher amounts of variance explained (Fig. EV1).

7. The lower explained variance in British African and South Asian groups is attributed partly to genetic differences. However, the sample sizes for these groups are small ($n=849$ and 823). Were sensitivity analyses (e.g., subsampling Europeans to match sample sizes) performed to confirm that the differences are not primarily due to reduced statistical power?

R2 response 7 We followed to helpful recommendation of the reviewer and implemented a subsampling strategy to complement our previously implemented strategy. Matching sample sizes thereby resulted in similar difference in the achieved explained variance as matching for the number of features found informative in the smaller ancestral strata (p6, lines 162-166).

8. The knowledge graph integrates self-reported medication data, which may be confounded by indication. Were comorbidities or concurrent treatments adjusted for in drug-protein associations? How robust are the inferred drug-target links?

R2 response 8 We apologize for the lack of description. All drug-protein links included in the knowledge graph were exclusively selected from the variance partition work that

represented a fully adjusted multivariable linear model accounting for relevant disease and drugs that may also affect the plasma levels of the protein of interest. We appreciate that self-reported drug intake is an imperfect surrogate of actual medication usage, and the cross-sectional design of our data might be prone to confounding by indication.

We further prioritized drug – (protein) target links only if there was coinciding support from independent genetic analysis. That is, only if a protein was associated with drug intake and the same protein was found to be life-long higher in individuals carrying genetic variants that mimic the effect of the drug on the target, we report those as engagement marker. We rephrased the corresponding section in the manuscript to illustrate more clearly the level of evidence provided in the graph (p10, lines 281-283) and acknowledge that our data-driven framework may not necessarily represent causal relationships (p13, lines 369-371).

Reviewer #3:

Summary:

The authors analyzed plasma proteins in UKBB individuals to identify factors responsible for variations in their plasmatic concentrations. They assessed numerous non-genetic features, patient's characteristics, protein-specific characteristics and previously identified genetic variants (cis- and trans-pQTLs), and tried to quantify the effect of all these variables on plasma protein levels, assessing such effects for each sex and across three different population of different ancestry. Next, the authors try to identify plasma proteins that could be used as disease biomarkers. The knowledge-graph provided by the authors is a great way to gather their results into an easy-to-use summary for large-scale studies.

We are grateful for the thoughtful comments by the reviewer and their appreciation of our work.

Considerations:

The work seems interesting, many results are in line with previous findings, but the overall writing is unprecise and make the whole work and the underlying methodology very opaque.

As of now, it is really hard for me to produce an clear and informed feedback about the scientific content of the manuscript, as the methods used are unclear, unprecise or not detailed in the manuscript. An additional writing effort from the authors is needed so that the quality of the scientific content can be vouched for.

Nevertheless, while I think this manuscript requires some changes to make it clear, the underlying hypotheses and questions appear sound and interesting. I'd like to be involved in a second round of review, once my following points will have be included.

Please note that readers will probably approach this work by reading the abstract & the results, and that i) a clearer description of what was previously done, including summary of relevant previous/referenced works when pointing toward a specific methodology/dataset, ii) explained, limited and consistent jargon usage, and iii) shorter sentences, would greatly improve the readability of this article.

- **R3 response 1** *We understand that the number of analyses done and results produced may present a challenge to some readers and we have now carefully revised the entire manuscript to address the I comments from this reviewer with respect to better accessibility to a broad audience. We kindly refer to that tracked changes in the revised manuscript instead of outlining the many textual changes here, for the sake of brevity (e.g., p3, lines 72-76; p7, lines 167-181; or p8, lines 215-225).*

For examples:

- This work is based on the field of population genetics, which uses the word "variation" in a very specific way, i.e. to design genetic variants.

While the authors used this word appropriately multiple times, it sometimes refer to plasma protein level variations: e.g. line 164: "The single strongest contributor to ancestral variation was the contribution of cis- and trans-pQTLs (Supplementary Tab. 7), likely explained by a combination of different reasons.", a sentence which also integrates references to genetic variations (cis- and trans- pQTLs) as contributors of the mentioned "ancestral variations", which can refer to both ancestral genetic variants, or to (what I believe the authors want to mention) ancestry-specific plasma protein explained variance. If so, please refer to these different concepts precisely, consistently and appropriately along the manuscript.

R3 response 2 *We agree with the reviewer that 'variation' is ambiguous and carefully revised its use in the new version of the manuscript, in addition to providing the relevant concepts (e.g., p7, lines 167-170).*

- Descriptors such as "ancestral regional lead variant" only make the reading more complex. Might be replaced with "lead genetic variant with ancestry-specific effect", which is indeed longer, but clearer and more precise. The "regional" descriptor is redundant with the "lead", as a lead variant is representative of several variants in a given region.

R3 response 3 *We carefully revised the text to increase precision in language, trying to balance length and content. We rephased the corresponding section accordingly (p7, line 171-172).*

- line 203 "To transcend statistical mapping of protein - disease associations that are unlikely to inform prioritisation of biologically or clinically relevant insights, we next built a framework designed to provide the foundation for robust biomarker identification in biobank-scale studies" -> sentence is too long, "prioritising insights" is very vague, "statistical mapping of [...] associations" doesn't make sense...

R3 response 4 *We simplified the complexity of sentences throughout the manuscript (e.g., p7, lines 194-195).*

Methods are incomplete, no mention of how QTLs score were obtained, while they are used throughout the manuscript.

R3 response 5 We apologize for the missing method sections and carefully revised the entire method section to provide the necessary detail to replicate the analyses done (e.g., p16, lines 447-449).

**** (non-exhaustive) **** corrections:

Results:

67-69 : "Protein targets thereby segregated into distinct clusters according to major influences on human health and but also pre-analytical variation activating platelets"?

R3 response 6 We rephrased the sentence for clarity (p5, lines 108-110).

140: extend -> extent

R3 response 7 We thank the reviewer for pointing out this typo.

151, 190,

R3 response 8 We are unsure to what the referee is referring to.

154: "differential or -specific" -> "differential or specific"

R3 response 9 We corrected the typo accordingly.

155: "and the sexes" - > "and sexes"

R3 response 10 We corrected the typo accordingly.

167-170: "We obtained evidence that differential effects of the same variant across ancestral groups most commonly accounted for differences and included 134 cis-pQTLs, almost all (n=130) of which could not be explained by different causal variants across ancestries." -> most commonly accounted for what differences? Sentence should be cut in half to improve readability, and the second part of the sentence reshaped to improve its clarity.

R3 response 11 We rephrased the sentence to improve readability according to the helpful suggestions by the reviewer (p7, line 178-181). Briefly, the sentence referred to the observation, that the same genetic variant explained significantly different amounts of variance across ancestries without being more or less common.

184: "paris" -> "pairs"

R3 response 12 We corrected the typo accordingly.

190: "ancestral-lead signals" -> ancestry-specific lead signal (ancestry-specific association signal between cis-pQTL and plasma protein levels)?

R3 response 13 *We corrected the typo accordingly.*

197: "trans-ancestral" -> "cross-ancestral", to avoid using confusion with trans-pQTLs which are also mentioned in the paper.

R3 response 14 *We appreciate the usage of 'trans' in a different context but prefer to keep the original wording to be in line with the general reporting of such findings in the field.*

198, 199: "ancestral-specific lead signal" -> ancestry-specific lead signal

R3 response 15 *We corrected the typo accordingly.*

205: "Among the 60,334 significant disease associations ($p < 4.4 \times 10^{-8}$) observed here and reported in other studies, we observed a more than 39-fold drop after regressing out foundations of plasma protein levels for 390 incident diseases)" -> "a 39-fold drop", and what are the foundations that are regressed out? No mention whatsoever of such "foundations" are made in the manuscript. A dedicated section in the Methods part of the manuscript is needed, detailing model used, included covariates... And why would you regress out all explaining covariates to identify biomarkers?

R3 response 16 *We understand that our previous reporting was insufficient and revised the introduction of the section accordingly to precisely report what was done (p7, line 194-203). We consider regressing out or accounting for covariates (which are statistically equivalent) of utmost importance for the identification of novel biomarkers, to ensure that they do not simply represent other, often readily measurable, participant characteristics. For example, the frequently associated and previously selected protein biomarker gastrin is likely an indicator of treatment with proton pump inhibitors that are frequently prescribed to people with multiple morbidities.*

For some participant characteristics proteins may still represent refined measures, such as CXCL17 for the adverse effects of smoking, and are hence of great value. However, accounting for as few as five tailored characteristics for respective protein levels resulted in a massive drop in association count demonstrating the need to consider the full spectrum of available data to identify novel biomarkers and potentially associated biology.

211: ">80% (n=1,300) showed considerable effect attenuation of {greater than or equal to}20%, suggesting residual confounding" -> 20% of what? residual confounding of what by what?

R3 response 16 *We appreciate that residual confounding is jargon specific to epidemiology and rephrased the sentence accordingly (p8, line 200-203). Briefly, we refer to the observation that effect estimates per 1 s.d. unit increase in protein biomarker levels were strongly attenuated once accounting for identified participant characteristics explaining plasma protein levels even if still formally meeting corrected statistical significance. Such effect attenuation is often a sign of incomplete adjustment, for example, when selected factors are themselves imperfect surrogate markers of the underlying reasons that links to the protein and to the disease creating a spurious association (the statistical definition of a confounder).*

222: "Less-established links with persistent considerable effect size included a 2.5-fold increased risk" -> persistent what?

R3 response 17 We apologize for the missing context. We rephrased the sentence to clarify that we refer to the comparison before and after accounting for selected participant characteristics that explained plasma protein levels in Cox-proportional hazard models (p8, line 212-215).

228: "We further obtained evidence for a potentially causal effect of high plasma levels of MMP12 on the risk of abdominal aortic aneurysm (hazard ratio=1.15; p-value=3.3x10⁻¹⁶ 230) by establishing a shared genetic signal (Posterior probability: rs17368814=97.5%, Supplementary Fig. 6)." -> Is it just not co-localization of signal? Claims of potential causality require more evidence, or at least a potential mechanistic explanation of causality.

R3 response 18 We strongly agree with the reviewer, that claims of causality should be supported by multiple lines of evidence. For MMP12, we propose exactly that and rephrased the section for clarity (p8/9, line 215-230).

Firstly, the association between plasma MMP12 levels and the onset of abdominal aortic aneurysm was almost unaffected after accounting for 32 factors that explained 43.2% of the variance in plasma MMP12 levels. Secondly, we provide evidence that the same genetic variant associated with life-long higher plasma MMP12 also associated with the risk of abdominal aortic aneurysm in independent cohorts. Finally, we reference experimental work supporting a role of MMP12 in AAA pathology, including pharmacological inhibition, but caution that such work has provided conflicting results and that more work is needed.

232: "Prioritisation of biomarker associations has previously been proposed using genetic imputation 14" -> what prioritisation of biomarker associations? The article referenced herein does not make any mention of prioritisation. Is it not only about identifying potential disease biomarkers among the assessed proteins?

R3 response 19 We apologize for the somewhat misleading introduction. Since the release of the 'omicspred' portal, we have witnessed attempts to propose replacing the measurement of plasma proteins solely with genetic imputation. This even led to (misguided) decision making by funding bodies. While such genetically anchored inference may help to prioritise potentially causal relationships, we caution here that they cannot replace measurements of dynamic protein readouts. A point important to make to funders, given the stark differences in cost between comparatively cheap genotyping and in-depth plasma proteomic profiling.

237: "This included not only strongly differing effect estimates, and hence power to stratify people at risk, but also discordant results whether considering cis- or trans pQTLs for imputing plasma protein levels " -> size effect estimates? What does "hence power to stratify people at risk" mean?

R3 response 20 As part of the general revisions to improve accessibility of the manuscript, we rephrased the section for clarity (p9, lines 239-240). Briefly, small effect sizes of pQTL variants in Cox proportional hazard models indicate that we cannot use those to reliably

distinguish between people at low or high risk to develop a given disease. In contrast, measuring either the corresponding or even a very different protein, might well be able to do so, as we demonstrated recently (Carrasco-Zanini et al., 2024, Nat. Med.).

Methods:

463-464: please briefly summarize how the pQTLs were mapped.

***R3 response 21** We apologize for the missing methods and added a description on how pQTLs were mapped and integrated in the present analyses (p15/16, lines 445-449).*

479: please briefly report the filters used.

***R3 response 22** We apologise, but we could not identify to which filters the referee is referring to.*

494: "We modelled categorical variables as random effects" -> why?

***R3 response 23** This part of the method section is now obsolete because of addressing similar statistical concerns raised by R2 (see response 2). In brief, we now use multivariable linear models and extract the partial explained variance as implemented in the `etasq()` function of the `heplots` R package to better account for the correlation structure among selected features.*

505-506: "This test was only done for variables not specific to only of the strata" -> a word is probably missing somewhere.

***R3 response 24** Thank you for spotting this error, that has now been corrected ('one' was missing).*

528: "silhouette coefficient" -> please briefly describe what it is.

***R3 response 25** The silhouette coefficient is a commonly applied method to determine the optimal number of clusters by comparing the distance of a point to its assigned cluster to all other cluster centres. Higher values indicate more coherent clustering. In the revised version of the manuscript, we have restrained from using Silhouette coefficients due to a higher dimensionality of the variance explained matrix and now rather use visual inspection of the UMAP plot alongside association analysis to determine meaningful instead of purely algorithmically defined clusters (p18, lines 518-525).*

535: "We declared ancestral-specific effects, if..." -> "We declared an effect as ancestry-specific if..."

***R3 response 26** We rephrased the sentence accordingly.*

539-540 "ancestries and computed the explained variance as well as tested for a significant interaction effects." -> interactions with what?

R3 response 27 We extended the description of this section to explain that we tested for a differential association between the selected variant and a given protein across ancestries (p18, lines 533-537). This is best done by including an interaction term in a linear model between ancestry and the variant of interest with the protein level as the dependent variable.

541 : "ancestral-differential explained variance" ->

R3 response 28 We rephrased the sentence for clarity.

557: "maximise the advantages of the knowledge graph" -> what advantages?

R3 response 29 We simplified the sentences to only report, that an interactive version of the knowledge had been generated. Advantages referred previously to the ability to examine subnetworks of particular interest by users interactively.

568: "Cox proportional hazards models" -> please briefly summarize what results such models produce

R3 response 30 Cox proportional hazard models are a statistical model for time-to-event data. In contrast to logistic regression that models the odds for a binary outcome to occur dependent on the variable of interest ('exposure'), Cox models additionally consider a temporal component by specifically modelling the time between the sample was taken and the onset of the disease. For example, participants experiencing an event only after ten years following the inclusion into the study are likely to have much lower plasma protein levels compared to those developing the disease already event after two years. Such information would be lost when, for example, using simpler logistic regression models.

572: "cis-pQTL scores, and trans-pQTL scores" -> how were they computed?

R3 response 31 We apologize for the missing information that has now been added to the manuscript (p16, lines 447-449). Briefly, pQTL scores were computed for each participant as weighted sum of protein increasing alleles. Weights were thereby obtained from the trans-ancestral meta-analysis performed by Sun et al. Nature 2023.

580: "Schoenfeld residuals" -> please briefly summarize what they are

R3 response 32 To model a time-dependent effect, Cox models assume a constant risk to experience an event over time, the proportional hazard assumption. Schoenfeld residuals can be used to diagnose departure from this proportionality and are computed for each individual experiencing an event at a time and covariate in the model as the deviation from the corresponding risk-weighted average of covariate values among all other participants at risk with a similar time to event. We clarified the reference in the text (p10, line 576)

586: "tissue-specifically expressed proteins" -> proteins with tissue-specific expression patterns

R3 response 33 *We rephrased the sentence following this helpful comment.*

587: "were enriched among those best explained by the same major biological factors."
"were enriched among the proteins best explained by the same major biological factors",
and what do you mean by "the same major biological factors"? Where is this similarity coming from?

R3 response 34 *We apologise for the ambiguities and rephrased the section accordingly (p20, lines 583-585). Briefly, for each participant characteristic, our approach revealed robustly associated lists of proteins. Each of those lists was tested for specific expression of related mRNA levels in tissues or cell-types.*

591 -> same question as for line 587

R3 response 35 *We expanded the explanation accordingly (p20, lines 595-597).*

Figures:

F1b: not referenced in the text.

R3 response 36 *We added the missing reference to the text (p4, line 84).*

F1c: very hard to read, categories are not distinguishable for each protein, not sure of the interest of this figure in the main body of this manuscript.

R3 response 37 *We appreciate the small scale and complexity of the figure but argue to retain it in the main text for multiple reasons. Firstly, it provides a transparent summary of the achieved explained variance across all protein targets. Secondly, the colour distribution across bars is indicative of the major factors explaining plasma protein levels. Lastly, it illustrates, as also outlined by reviewer 2, the gap of knowledge still existing in explaining variance in plasma protein measurements.*

F1d : omingenic -> omnigenic

R3 response 38 *We corrected the typo accordingly.*

SF1a: "darker colours indicate features passing statistical significance" -> cant see any darker colour?

R3 response 39 *We replaced the colour shades with a simple line separating significant from non-significant attributes.*

SF1a: the authors assess the explained variance of plasma proteins using protein characteristics extracted and/or computed from UniProt & the Human Protein Atlas. I'm not sure that

***R3 response 40** We are unsure about what the reviewer is referencing to here, but the categories have been selected based on the feature selection algorithm. We appreciate, that some of those are likely to be correlated, and that we cannot be certain about the truly underlying factors.*

SF2c-i : please homogenize the colorbar used for the explained variance, as it changes with each plot, it would improve the readability of the figure.

***R3 response 41** We appreciate that a standardised colour gradient would improve comparability but deliberately decided against it due to the vastly different amount of maximal variance explained by some of the factors. The plots serve mainly to support the clustering assignment, so the need to highlight proteins explained to a certain degree by each of the selected features.*

SF3e -> cant see any darker colour?

***R3 response 41** We are sorry, but we cannot locate the figure the reviewer is referring to.*

SF4e: not referenced in text

***R3 response 42** We apologize for the oversight and added the reference to the text (page 7, line 191).*

10th Sep 2025

Manuscript Number: MSB-2025-13038R

Title: Machine learning-guided deconvolution of plasma protein levels

Dear Dr. Pietzner,

Thank you for the submission of your revised manuscript to Molecular Systems Biology. I am pleased to inform you that we will be able to accept your manuscript pending the following final amendments and appropriate response to reviewers:

1) Please include a README file on Github with practical use instructions for potential future users of your code.
2) Please specify author contributions in our submission system. CRediT has replaced the traditional author contributions section because it offers a systematic machine-readable author contributions format that allows for more effective research assessment. You are encouraged to use the free text boxes beneath each contributing author's name to add specific details on the author's contribution. More information is available in our guide to authors:

<https://www.embopress.org/page/journal/17574684/authorguide#authorshipguidelines>

3) In the manuscript, please correct the reference citation in the reference list to be alphabetical (not numerical). Where there are more than 10 authors on a paper, only the first 10 should be listed, followed by "et al.". DOIs should only be used for preprints and datasets that have not been published yet. Please check "Author Guidelines" for more information.

<https://www.embopress.org/page/journal/17574684/authorguide#referencesformat>

4) In the Methods, please take care of the following:

- The Materials and Methods section should be renamed to "Methods".

- BioRender should be acknowledged at the end of the Methods section in the following way:

Graphics:

(some of the... OR Figure #... OR synopsis) Graphics were created with BioRender.com.

- Currently you have included multiple sections in the Methods that should be clearer on what you have done for the current study - e.g.

'Study participants' - unless you recruited the participants for the UKB cohort and received ethical approval, this information should be limited to a citation and your specific inclusion criteria for the subset of participants that you analyzed. The ethics approval and Application numbers should be removed and it should be clarified to exactly what you did with the information. The Author Checklist should also be updated to indicate that the present study does not include Human Participants. However, if you played a role in participant recruitment, please state that the experiments conformed to the principles set out in the WMA Declaration of Helsinki and the Department of Health and Human Services Belmont Report. Please note that this is a separate statement from the specific ethics committee approval and informed consent.

'proteomics measurements' - unless you performed the proteomics measurements in the study, this information should also be limited and clarified to what exactly was performed in the study, e.g. if there were specific criteria or measurements included in your analysis you could indicate that you 're-analyzed publicly available data from [XXX]'. The summary of the process is not necessary unless you have performed these methods. However, if the proteomics were performed, the datasets need to be made available in a ProteomeXchange repository and the Reagents and Tools table needs to be updated.

'Genotyping and ancestral assignment' - if the UK Biobank samples were genotyped previously, please remove these details on the microarrays and ensure that only details of what you did with the information are included. If the microarrays were experimentally performed by you, you may leave in this information, but will need to make the datasets publicly available and the Reagents and Tools table will need to be updated.

- Although you have indicated in the Author Checklist that a statement on whether or not blinding was done is included in the Methods, we were not able to locate this statement.

5) Please place individual sections of the manuscript in the following order: Title page - Abstract & Keywords - Introduction - Results - Discussion - Methods - Data Availability - Acknowledgements - Disclosure and Competing Interests Statement - References - Figure Legends - Expanded View Figure Legends.

6) For the figures and figure legends, please take care of the following:

- The callouts for Fig. 1e and Fig. 3e are missing in the main manuscript.

- Please note that the exact p values are not provided in the legends of figures 4A-D

- Please indicate the statistical test used for data analysis in the legends of figures 4A-D; 5C, EV4 D, E; EV5, EV6, EV7

- Please note that the box plots need to be defined in terms of minima, maxima, centre, bounds of box and whiskers, and percentile in the legends of figures EV1 A, C, D, F, G, H; EV4 C

- Please note that information related to n is missing in the legends of figures 1E, 5B, EV1 A, C, D, F, G, H; EV4 C

- Please note that the error bars are not defined in the legend of figure 5B.

7) Please upload each EV table as one .xsl file per table and rename them to Dataset EV1-EV15. Each dataset will need its legend added to the corresponding file in a separate tab. Please also be sure to include a callout for each file in the main manuscript text - currently only Table EV7 has a callout. The nomenclature should be corrected in all places: source file names, titles in the system, legends in each file.

8) Please ensure that all funding sources are entered into the manuscript submission system. Currently the following are missing in the manuscript submission system: DZHK; BMBF grant numbers: 031A533A, 031A533B, 031A534A, 031A535A, 031A537A,

031A537B, 031A537C, 031A537D, 031A538A

9) Please use the passive voice for the synopsis text (i.e. remove the 'we' in the standfirst and bullet point). Please check your synopsis text and image before submission with your revised manuscript. Please be aware that in the proof stage minor corrections only are allowed (e.g., typos).

10) As part of the EMBO Publications transparent editorial process initiative (see our policy here: https://www.embopress.org/transparent-process#Review_Process), Molecular Systems Biology will publish online a Peer Review File (PRF) to accompany accepted manuscripts. This file will be published in conjunction with your paper and will include the anonymous referee reports, your point-by-point response and all pertinent correspondence relating to the manuscript. Let us know whether you agree with the publication of the PRF and as here, if you want to remove or not any figures from it prior to publication. Please note that the Authors checklist will be published at the end of the PRF.

11) After your paper is published, we may promote it on social media. If you have any handles or hashtags for Bluesky you would like included, please let us know.

12) Please provide a point-by-point letter INCLUDING my comments as well as the reviewer's reports and your detailed responses (as Word file).

I look forward to reading a new revised version of your manuscript as soon as possible.

Yours sincerely,

Poonam Bheda, PhD
Scientific Editor
Molecular Systems Biology

Reviewer #1:

I thank the authors for their additional analyses and have no more comments.

!

Reviewer #2:

I believe the authors have done an excellent job addressing my comments, and the manuscript has improved substantially since initial submission. I have no further feedback to provide.

Reviewer #3:

Generalities:

I thank the authors for their exhaustive answers to my questions, and I salute the additional work performed, especially regarding the Methods and the replication analysis.

Minor comments:

Fig 1d: where are the labels of the clusters coming from (e.g. monogenic, phenotypic diverse...)? They are quite unrelated (monogenic vs platelet...), or seem unjustified? (e.g. the "liver/macrophage" cluster). Would be interesting to get more information on the provenance of the labels, and a few words informing the reader about their relevance. As these labels are used throughout the paper, I think it is important to define them properly.

Fig5b: Might ease readability if the legends from fig1c are also indicated here, as I assume that the color codes are the same for both figures?

Section "Protein biomarker discovery and pruning for incident diseases" (l 195):

I'm not sure I'm understanding clearly the opinion of the authors: Is the observed attenuation problematic or not? At first, it seems that the authors present it as potentially problematic and suggest that suggest caution when correction for associated characteristics, as it could dampen the signal, but then use these attenuated results for the rest of the section?

In addition, the method section for the imputation of plasma protein levels from genotyping data is missing. If the imputation was not performed for this work, please point toward the source of these data.

If the authors do not consider that the observed attenuation is problematic:

What if the characteristics explaining variation in plasma proteins, that were regressed out of the associated plasma protein levels, are not explanatory factors but actually co-varying with the protein of interest or even co-implicated in the disease? I'm not sure that correcting for all characteristics that correlate with a given plasma protein expression levels systematically improves the analysis, as it is likely that the manifestation of a disease is likely to eventually impact the levels of many proteins, which all are and should be associated with the corresponding disease?

"For example, the five most significant genetically proxied protein - disease association were not among the twenty most strongly associated measured protein - disease associations for two-thirds of all diseases considered (268 out of 419)". Why only two-third of all diseases were considered? In addition, the "Protein - incident disease analyses" section in the Methods refer to 390 diseases?

typos:

74 -> "characteristics characteristics" -> either "characteristics" or "characteristics of characteristics"

102 -> generalisability

175 -> "arylsulfatase A" -> "arylsulfatase A (ARSA)" so that it is easier to locate the corresponding protein in the associated plots.

204 -> preciser -> more precise

205 -> characterisitcs -> characteristics

339 -> characterisitcs -> characteristics

455 -> gentic -> genetic

486 -> by artifical introducing -> by introducing 10 artificial random...

532 -> characterisitcs -> characteristics

537 -> declared and effect -> declared an effect

EDITORIAL COMMENTS

1) Please include a README file on Github with practical use instructions for potential future users of your code.

Response 1 We added a README file to our GitHub repository as suggested.

2) Please specify author contributions in our submission system. CRediT has replaced the traditional author contributions section because it offers a systematic machine-readable author contributions format that allows for more effective research assessment. You are encouraged to use the free text boxes beneath each contributing author's name to add specific details on the author's contribution. More information is available in our guide to authors:

<https://www.embopress.org/page/journal/17574684/authorguide#authorshipguidelines>

Response 3 We specified author contributions in the submission system.

3) In the manuscript, please correct the reference citation in the reference list to be alphabetical (not numerical). Where there are more than 10 authors on a paper, only the first 10 should be listed, followed by "et al.". DOIs should only be used for preprints and datasets that have not been published yet. Please check "Author Guidelines" for more information.

<https://www.embopress.org/page/journal/17574684/authorguide#referencesformat>

Response 3 We adopted the suggested citation style.

4) In the Methods, please take care of the following:

- The Materials and Methods section should be renamed to "Methods".

Response 4.1 We renamed the section accordingly.

- BioRender should be acknowledged at the end of the Methods section in the following way:

Graphics:

(some of the... OR Figure #... OR synopsis) Graphics were created with BioRender.com.

Response 4.2 We acknowledge BioRender.com as suggested.

- Currently you have included multiple sections in the Methods that should be clearer on what you have done for the current study - e.g.

'Study participants' - unless you recruited the participants for the UKB cohort and received ethical approval, this information should be limited to a citation and your specific inclusion criteria for the subset of participants that you analyzed. The ethics approval and Application numbers should be removed and it should be clarified to exactly what you did with the information. The Author Checklist should also be updated to indicate that the present study does not include Human Participants. However, if you played a role in participant recruitment, please state that the experiments conformed to the principles set out in the WMA Declaration of Helsinki and the Department of Health and Human Services Belmont Report. Please note that this is a separate statement from the specific ethics committee approval and informed consent.

Response 4.3 We rephrased the section accordingly to reflect that we have not been involved in participant recruitment (p15, lines 388-394).

'proteomics measurements' - unless you performed the proteomics measurements in the study, this information should also be limited and clarified to what exactly was performed in the study, e.g. if there were specific criteria or measurements included in your analysis you could indicate that you 're-analyzed publicly available data from [XXX]'. The summary of the process is not necessary unless you have performed these methods. However, if the proteomics were performed, the datasets need to be made available in a ProteomeXchange repository and the Reagents and Tools table needs to be updated.

Response 4.4 We followed the editorial recommendations and rephrased the section to clearly state that we have not been involved in data generation (p15, lines 396-402).

'Genotyping and ancestral assignment' - if the UK Biobank samples were genotyped previously, please remove these details on the microarrays and ensure that only details of what you did with the information are included. If the microarrays were experimentally performed by you, you may leave in this information, but will need to make the datasets publicly available and the Reagents and Tools table will need to be updated.

Response 4.5 We followed the editorial recommendations and rephrased the section to clearly state that we have not been involved in data generation (p16, lines 457-460).

- Although you have indicated in the Author Checklist that a statement on whether or not blinding was done is included in the Methods, we were not able to locate this statement.

Response 4.6 We added a statement that researchers were not blinded to the outcome variables to facilitate downstream analysis (p19, lines 517-518).

5) Please place individual sections of the manuscript in the following order: Title page - Abstract & Keywords - Introduction - Results - Discussion - Methods - Data Availability - Acknowledgements - Disclosure and Competing Interests Statement - References - Figure Legends - Expanded View Figure Legends.

Response 5 We arranged the sections of the manuscript accordingly.

6) For the figures and figure legends, please take care of the following:
- The callouts for Fig. 1e and Fig. 3e are missing in the main manuscript.

Response 6.1 We corrected the call outs in the revised manuscript, which we missed to updated during the previous round of review.

- Please note that the exact p values are not provided in the legends of figures 4A-D

Response 6.2 We expanded the figure legends to clarify that the reported p-values represent the multiple testing corrected threshold to declare statistical significance and refer to Dataset EV10 for exact association statistics.

- Please indicate the statistical test used for data analysis in the legends of figures 4A-D; 5C, EV4 D, E; EV5, EV6, EV7

Response 6.3 We added the relevant information to the figure legends but note that some were already given largely referring to Cox-proportional hazard or linear regression models.

- Please note that the box plots need to be defined in terms of minima, maxima, centre, bounds of box and whiskers, and percentile in the legends of figures EV1 A, C, D, F, G, H; EV4 C

Response 6.4 We added the relevant description to the figure legends.

- Please note that information related to n is missing in the legends of figures 1E, 5B, EV1 A, C, D, F, G, H; EV4 C

Response 6.5 We added the relevant numbers to the figure legends.

- Please note that the error bars are not defined in the legend of figure 5B.

Response 6.6 We added information on how errors bars were derived.

7) Please upload each EV table as one .xsl file per table and rename them to Dataset EV1-EV15. Each dataset will need its legend added to the corresponding file in a separate tab. Please also be sure to include a callout for each file in the main manuscript text - currently only Table EV7 has a callout. The nomenclature should be corrected in all places: source file names, titles in the system, legends in each file.

Response 7 We prepared the EV tables as suggested and adopted callouts in the main text accordingly.

8) Please ensure that all funding sources are entered into the manuscript submission system. Currently the following are missing in the manuscript submission system: DZHK; BMBF grant numbers: 031A533A, 031A533B, 031A534A, 031A535A, 031A537A, 031A537B, 031A537C, 031A537D, 031A538A

Response 8 We added all grant information into the submission system.

9) Please use the passive voice for the synopsis text (i.e. remove the 'we' in the standfirst and bullet point). Please check your synopsis text and image before submission with your revised manuscript. Please be aware that in the proof stage minor corrections only are allowed (e.g., typos).

Response 9 We carefully revised the synopsis to be written in passive voice.

10) As part of the EMBO Publications transparent editorial process initiative (see our policy here: https://www.embopress.org/transparent-process#Review_Process), Molecular Systems Biology will publish online a Peer Review File (PRF) to accompany accepted manuscripts. This file will be published in conjunction with your paper and will include the anonymous referee reports, your point-by-point response and all pertinent correspondence relating to the manuscript. Let us know whether you agree with the publication of the PRF and as here, if you want to remove or not any figures from it prior to publication. Please note that the Authors checklist will be published at the end of the PRF.

Response 10 We agree and support publication of the PRF and do not wish to omit any figures prior publication.

11) After your paper is published, we may promote it on social media. If you have any handles or hashtags for Bluesky you would like included, please let us know.

Response 11 None of the authors or institutions is active on Bluesky.

12) Please provide a point-by-point letter INCLUDING my comments as well as the reviewer's reports and your detailed responses (as Word file).

Response 12 A point-by-point letter including editorial and reviewer comments is provided.

REVIEWER COMMENTS

Reviewer #1:

I thank the authors for their additional analyses and have no more comments.

!

R1 response 1 We thank the reviewer for the dedicated time and constructive feedback that improved our manuscript.

Reviewer #2:

I believe the authors have done an excellent job addressing my comments, and the manuscript has improved substantially since initial submission. I have no further feedback to provide.

R2 response 1 We thank the reviewer for the positive response, dedicated time and constructive feedback that improved our manuscript.

Reviewer #3:

Generalities:

I thank the authors for their exhaustive answers to my questions, and I salute the additional work performed, especially regarding the Methods and the replication analysis.

R3 response 1 We thank the reviewer for the positive response, dedicated time and constructive feedback that improved our manuscript.

Minor comments:

Fig 1d: where are the labels of the clusters coming from (e.g. monogenic, phenotypic diverse...)? They are quiet unrelated (monogenic vs platelet...), or seem unjustified? (e.g. the "liver/macrophage" cluster). Would be interesting to get more information on the provenance of the labels, and a few words informing the reader about their relevance. As these labels are used throughout the paper, I think it is important to define them properly.

R3 response 2 We followed the helpful advice of the reviewer and now clearly outline, how labels were derived (p5, lines 109-110, and p18/19, lines 511-516).

Fig5b: Might ease readability if the legends from fig1c are also indicated here, as I assume that the color codes are the same for both figures?

R3 response 3 We added the colour code to Figure 5b as suggested.

Section "Protein biomarker discovery and pruning for incident diseases" (l 195):

I'm not sure I'm understanding clearly the opinion of the authors: Is the observed attenuation problematic or not? At first, it seems that the authors present it as potentially problematic and suggest that suggest caution when correction for associated characteristics, as it could dampen the signal, but then use these attenuated results for the rest of the section?

In addition, the method section for the imputation of plasma protein levels from genotyping data is missing. If the imputation was not performed for this work, please point toward the source of these data.

If the authors do not consider that the observed attenuation is problematic:

What if the characteristics explaining variation in plasma proteins, that were regressed out of the associated plasma protein levels, are not explanatory factors but actually co-varying with the protein of interest or even co-implicated in the disease? I'm not sure that correcting for all characteristics that correlate with a given plasma protein expression levels systematically improves the analysis, as it is likely that the manifestation of a disease is likely to eventually impact the levels of many proteins, which all are and should be associated with the corresponding disease?

"For example, the five most significant genetically proxied protein - disease association were not among the twenty most strongly associated measured protein - disease associations for two-thirds of all diseases considered (268 out of 419)". Why only two-third of all diseases were considered? In addition, the "Protein - incident disease analyses" section in the Methods refer to 390 diseases?

R3 response 4 We appreciate that this section still benefited from clarification, and we carefully revised the corresponding section accordingly (p8, lines 193-208). In essence, this section aims to identify protein biomarkers that are superior to other participant characteristics in predicting disease onset. Substantial effect attenuation, irrespective of statistical significance, is thereby a sign of imperfect measurement of the truly underlying factor that connects plasma protein levels with disease onset. In other words, the protein biomarkers appear only superior because other characteristics are imprecisely measured (e.g., quantifying the precise amount of alcohol people are consuming instead of self-reported categories of drinking behaviour). Most importantly, proteins persistently

associated with disease risk could not be explained by the >1,800 factors considered and hence are most promising to take forward for further (costly) evaluation.

We note that some of the adjustments may indeed represent potentially causal pathways, by which the covariate acts on disease risk via plasma protein levels. However, this still implies that measuring the covariate is sufficient to monitor disease risk.

We further describe computation of imputed protein levels in the Methods section (p16, lines 439-444) and corrected the typo when referring to the number of diseases in the main text.

typos:

74 -> "characteristics characteristics" -> either "characteristics" or "characteristics of characteristics"

102 -> generalisability

175 -> "arylsulfatase A" -> "arylsulfatase A (ARSA)" so that it is easier to locate the corresponding protein in the associated plots.

204 -> preciser -> more precise

205 -> characterisitcs -> characteristics

339 -> characterisitcs -> characteristics

455 -> gentic -> genetic

486 -> by artifical introducing -> by introducing 10 artificial random...

532 -> characterisitcs -> characteristics

537 -> declared and effect -> declared an effect

R3 response 5 *We thank the reviewer for the careful reading and pointing out those typos.*

22nd Sep 2025

Manuscript number: MSB-2025-13038RR

Title: Machine learning-guided deconvolution of plasma protein levels

Dear Dr. Pietzner,

Congratulations on an excellent manuscript, I am pleased to inform you that your manuscript has been accepted for publication in Molecular Systems Biology. Thank you for your comprehensive response to referee concerns. It has been a pleasure to work with you to get this to the acceptance stage.

Yours sincerely,

Sincerely,

Poonam Bheda, PhD
Scientific Editor
Molecular Systems Biology
